

# How well do hydrological models simulate streamflow extremes and drought-to-flood transitions?

Eduardo Muñoz-Castro[1,2,3], Bailey J. Anderson[1,2,3], Paul C. Astagneau[1,2,3], Daniel L. Swain[4,5,6], Pablo A. Mendoza[7,8], and Manuela I. Brunner[1,2,3]

[1]WSL Institute for Snow and Avalanche Research SLF, Davos Dorf, Switzerland
[2]Climate Change, Extremes and Natural Hazards in Alpine Regions Research Center CERC, Davos Dorf, Switzerland
[3]Institute for Atmospheric and Climate Science, ETH Zurich, Zurich, Switzerland
[4]California Institute for Water Resources, University of California Agriculture and Natural Resources, Davis, CA, USA
[5]Institute of the Environment and Sustainability, University of California, Los Angeles, Los Angeles, CA, USA
[6]Capacity Center for Climate and Weather Extremes, National Center for Atmospheric Research, Boulder, CO, USA
[7]Civil Engineering Department, Universidad de Chile, Santiago, Chile
[8]Advanced Mining Technology Centre (AMTC), Universidad de Chile, Santiago, Chile

**Correspondence:** Eduardo Muñoz-Castro (eduardo.munoz-castro@slf.ch)

**Abstract.** The impacts of floods can be enhanced when they occur shortly after drought. Models can be a useful tool to better understand the processes and mechanisms driving the response of floods occurring in close succession to streamflow drought. However, it is yet unclear how well hydrologic models capture these compound extreme events and which modeling decisions are most important for high model performance. To address this research gap, we calibrated four conceptual bucket-type hydrological models with different structures (GR4J, GR5J, GR6J, and TUW) for 63 catchments in Chile and Switzerland using different calibration strategies. We tested different configurations of the Kling-Gupta efficiency (KGE) formulation for model calibration to assess their performance in simulating and detecting observed transitions. We assessed the relative importance of different methodological choices including model structure, streamflow transformation, and KGE formulation and weights. We demonstrate that model performance as expressed by the KGE or NSE does not guarantee a good performance in terms of detecting streamflow extremes and their transitions. Furthermore, we show that a model's performance with respect to capturing extreme events primarily depends on how well it captures streamflow timing (i.e., correlation between observations and simulations) rather than other hydrological signatures or variables such as evapotranspiration or snow water equivalent. Our results also highlight that model structure and catchment characteristics as well as meteorological forcing play a key role in the detection of transitions. Specifically, we demonstrate that drought-to-flood transitions are more difficult to capture in semi-arid high-mountain catchments than in humid low-elevation catchments. Finally, our study provides guidelines for further model improvements with respect to drought-to-flood transitions, which can support process understanding related to these compound events, identifying regions prone to this type of event, and contribute to improved risk management – aspects that will enhance preparedness.





## 1 Introduction

Hydrological extreme events such as streamflow droughts and floods are expected to become more frequent, severe, and persistent in a warming climate (e.g., Gu et al., 2023; Asadieh and Krakauer, 2017; Martin, 2018; Tabari et al., 2021), with severe impacts on infrastructure, agriculture, water supply, and hydropower generation (e.g., McClymont et al., 2020; McMartin et al., 2018; Lehner et al., 2006; Sivakumar, 2011; Wasti et al., 2022), as well as social and political systems (e.g., Doocy et al., 2013; Hurlbert and Gupta, 2017; Kiem and Austin, 2013; Visconti, 2022). Studies focusing on hydrological extreme events and their impacts often assume temporal and/or spatial independence between them, neglecting that extremes may be multivariate

phenomena (Banfi and De Michele, 2022; Brunner, 2023). However, the impacts of floods can be enhanced when they occur during or shortly after a streamflow drought (e.g., Barendrecht et al., 2024; Swain et al., 2018; He and Sheffield, 2020; Rashid and Wahl, 2022). For instance, Handwerger et al. (2019) and Valenzuela et al. (2022) have demonstrated an increase in the occurrence of landslides in California and Chile due to shifts from drought to intense precipitation. Similarly, Dietze et al. (2022) have shown that the 2018-2020 drought in Europe has enhanced the debris mobilisation during the 2021 flood in the Eifel

region of western Germany and Belgium. In 2017, intense precipitation broke the 2012-2016 drought in California and led to severe flooding, the activation of the emergency spillway of the Lake Oroville dam for the first time in its history, and the declaration of emergency (Griffin and Anchukaitis, 2014; Robeson, 2015; Wang et al., 2017). These examples highlight the need to integrate droughts and floods within the same analysis framework (e.g., Ward et al., 2020; Quesada-Montano et al., 2018; Di Baldassarre et al., 2017). Despite this, droughts and floods have been mostly studied as independent events. As a result, the

occurrence and drivers of drought-to-flood transitions are not completely understood (e.g., Matanó et al., 2022, 2024; Brunner, 2023; Götte and Brunner, 2024). Model experiments are needed to explore the sensitivity of flood properties to drought occurrence and characteristics.

Hydrological models are useful tools for studying how streamflow - and/or other hydrological fluxes and states - react

to variations in different meteorological and environmental inputs. Models enable insight into the mechanisms which drive catchment response to variations, through their representation of hydrological processes (Hrachowitz and Clark, 2017). In recent decades, efforts have been made to improve hydrological modeling in terms of the representation of spatial variability (e.g., Dembélé et al., 2020), the simulation of low (e.g., Garcia et al., 2017) and high-flows (e.g., Mizukami et al., 2019), or the representation of flood-triggering mechanisms and spatiotemporal coherence (e.g., Brunner et al., 2020, 2021), under

current and changing climate conditions (e.g., Fowler et al., 2018). However, modeling hydrological extreme events such as droughts and floods is still very challenging (e.g., Mizukami et al., 2019; Bruno et al., 2024). This can be particularly difficult in contexts where multiple variables are of interest, e.g. when modeling the dependence between flood peaks and volumes (Brunner and Sikorska-Senoner, 2019) or the spatial dependence of floods happening in different locations (Brunner et al., 2021). This complexity suggests that capturing consecutive drought-to-flood events might not be trivial either. As model

evaluations targeted at compound extremes have not yet been performed, it is still unclear how well hydrological models can in fact capture drought-to-flood transitions.





Hydrological modeling involves different choices about model structure (i.e., process representation and parametrization), spatial discretization, meteorological forcing, and calibration approach (e.g., period for calibration/evaluation, hydrological target variables or signatures used in objective function), which affect hydrological simulations and whose importance might
vary depending on the modeling purpose (e.g., Mendoza et al., 2016; Mizukami et al., 2016; Baez-Villanueva et al., 2021; Guo et al., 2017; Melsen et al., 2019). Previous studies have highlighted that such modeling decisions can substantially influence simulated hydrological extremes and their uncertainties (e.g., Alexander et al., 2023; Melsen and Guse, 2019; Melsen et al., 2019). They have also shown that among these choices, the choice of objective function for model calibration, model structure, and spatial discretization (forcings and domain) are the most influential decisions on modeling outcomes. While these previous
studies have focused on analyzing the impacts of modeling decisions on drought and flood attributes (e.g., severity, duration), they have not looked at how these decisions influence event detection, i.e. whether or not the model can capture extreme events below or above a certain threshold. Furthermore, these previous studies have focused on individual extremes instead of looking at them in a multivariate setting. As such, it is yet unclear how individual modeling decisions might influence the representation of hydrological transitions.

Hydrological modeling often relies on model calibration to define parameters that minimize discrepancies between observations and simulations of a target variable (e.g., streamflow). Calibration uses an objective function to measure the similarity between observations and simulations. In general, these objective functions are defined based on "least squares" formulations such as the widely used Nash-Sutcliffe Efficiency (NSE; Nash and Sutcliffe, 1970) and Kling-Gupta Efficiency (KGE; Gupta et al., 2009). However, alternative objective functions have been suggested to improve the robustness of calibrated parameters.
This is especially relevant in the context of a changing climate (e.g., Fowler et al., 2018), and might improve hydrological consistency through metrics that better capture the statistical and dynamic properties of hydrological time series (referred in the literature as hydrological signatures; e.g., Yilmaz et al., 2008; Shafii and Tolson, 2015; McMillan, 2021). While these alternatives are available and concerns regarding the standard options (e.g., Clark et al., 2021; Knoben et al., 2019; Cinkus et al., 2023) have been noted, KGE and NSE continue to be widely used for model calibration and evaluation (e.g., Klemeš, 1986;
Motavita et al., 2019; Seibert et al., 2019; Beven, 2025).

The Kling-Gupta Efficiency (KGE), originally proposed by Gupta et al. (2009), is one of the most popular performance metrics used in hydrology. KGE is more widely used than other well-known metrics, such as the NSE, because of its interpretability and the diagnostic power provided, thanks to the possibility of disaggregating it into its three components, namely bias, variability, and correlation. It has been applied for many modeling purposes, including studies focused on streamflow
extremes (e.g., Gu et al., 2023; Hirpa et al., 2018). In these studies, calibrations are considered successful if the KGE performance exceeds a certain threshold during both the calibration and evaluation periods (e.g., KGE > 0.4). However, there is often no explicit evaluation of how drought or flood events are represented at the event scale. Thus, the suitability of KGE - and alternative formulations (Gupta et al., 2009; Kling et al., 2012; Pool et al., 2018; Tang et al., 2021; Pizarro and Jorquera, 2024) and adaptations (e.g., transformations and weights; Garcia et al., 2017; Wu et al., 2025; Mizukami et al., 2019) – for calibrat-
ing models aimed at studying streamflow extreme events, and consecutive extremes in particular, has not yet been sufficiently evaluated.





In summary, there exist significant research gaps related to modeling floods which occur in close succession following droughts, that is, drought-to-flood transitions. In particular, it is unclear how different modeling decisions - such as the choice of hydrological model, objective function, and streamflow transformations - affect the simulation of transition events. Furthermore, it remains to be explored which modeling choices are most suitable for capturing these compound hydrological extreme events without compromising hydrological consistency (i.e., representation of different hydrological processes or properties). To address these research gaps, our study aims to investigate the extent to which hydrological models can represent consecutive drought-to-flood transitions and the impact of model complexity and calibration choices on their representation. Specifically, we investigate the following research questions:

– How suitable is the KGE for calibrating models aimed at jointly simulating streamflow droughts and floods?

– Which KGE formulations, weights, and streamflow transformations are most suitable for simulating drought-to-flood transitions?

– To what extent does model structure influence the detection and representation of streamflow transitions?

– Which are the key hydrological processes that have to be captured by models to well simulate drought-to-flood transitions?

To address these questions, we first calibrate four conceptual bucket-type hydrological models (GR4J, GR5J, GR6J, and TUW) for 63 diverse catchments in Chile and Switzerland. For this calibration experiment, we tested different configurations of the Kling-Gupta efficiency (KGE) - listed in Table 1 - to assess their performance in simulating and detecting observed transitions. These configurations included five KGE formulations (Table 1), two streamflow transformations (i.e., Q and 1/Q) and their linear combination (i.e., 0.5*KGE(Q)+0.5*KGE(1/Q)), and four different weights applied to the variability term of the KGE ($c_2$=1,2,4,8). Second, we assess the relative importance of each of these methodological choices for the detection of events and ensuring hydrological consistency. Finally, we explore the link between model performance and the representation of different hydrological fluxes and states during transition events.

This comprehensive assessment of the influence of different modeling decisions on the accuracy of drought-to-flood transition simulations will enable recommendations on potential future model improvements.

## 2 Study domain and data

The study domain encompasses 24 and 39 near-natural catchments in Chile (CL; Figure 1a) and Switzerland (CH; Figure 1b), respectively. These catchments are selected based on the availability of complete daily streamflow records between 1981 and 2020 for at least 30 years, with a complete year being defined as one in which all months had information for at least 90% of the days. The selected catchments span a wide range of hydroclimatic characteristics (Figure 1c), from energy to water-limited, and different hydrological regimes (Figure 1d), from snowmelt (e.g., p-seasonality < -0.5 and q-seasonality > 0.5) to rainfall-dominated (e.g., p-seasonality < -0.5 and q-seasonality < -0.5). Some catchments are placed above the water limit (i.e., Q/P





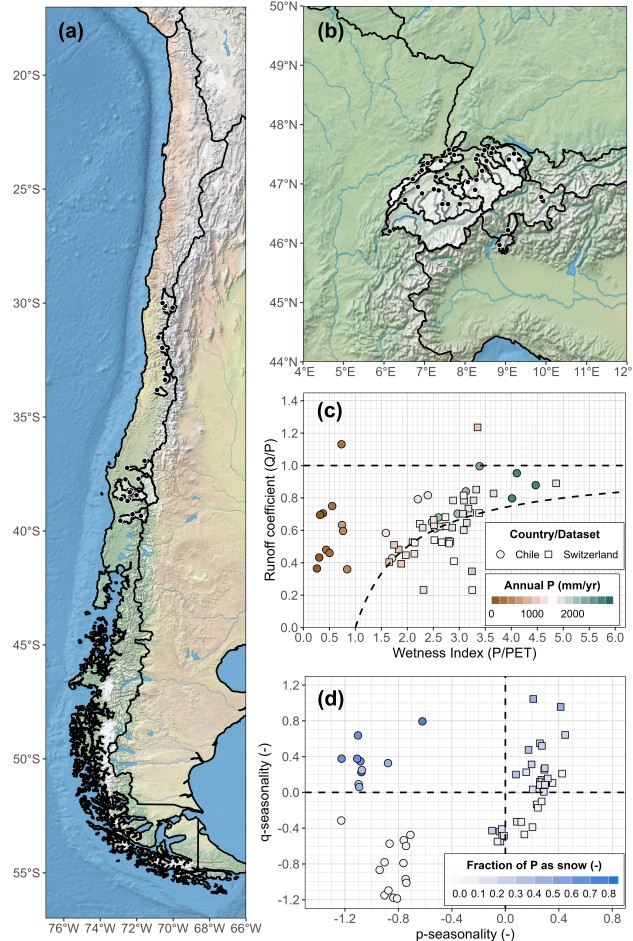

**Figure 1.** Study domain and hydroclimatic characteristics computed for the period 1981-2020 using data retrieved from CAMELS Chile (CL) and Switzerland (CH). Location of catchments across the study domain in (a) Chile and (b) Switzerland, (c) relationship between wetness index (PET/P), runoff coefficient (Q/P), and mean annual precipitation, and (d) relationship of seasonality of precipitation and streamflow and fraction of precipitation falling as snow. For p-seasonality and q-seasonality, values near 1 (-1) suggest strongly summer (winter) dominated precipitation or streamflow, while values close to zero indicate a uniform distribution across the year.

= 1) or below the energy limit (i.e., Q/P = 1 - 1/(P/PET); Figure 1c), which suggests an underestimation of precipitation or a surplus of streamflow. Consequently, there is a need to adjust the meteorological forcings, e.g. by correcting for precipitation
undercatch (e.g., Newman et al., 2015; Stisen et al., 2012; Hughes et al., 2021).

The CAMELS Chile (CL; Alvarez-Garreton et al., 2018a) and Switzerland (CH; Höge et al., 2023a) datasets are used to obtain the meteorological forcings, streamflow records, snow water equivalent (SWE) estimates, and catchment outlines for the catchments in the study domain. The meteorological forcings of both datasets, CR2Met version 2.5 for Chile (Boisier, 2023) and RhiresD version 2 for Switzerland (MeteoSwiss, 2023), are based on local gridded observation-based products, while SWE





products are based on a snow cover model and data assimilation (for more detail refer to Cortés and Margulis, 2017; Magnusson et al., 2014). We prefer these local products over global ones such as ERA5 (Hersbach et al., 2020) because of their reliance on observations and high spatial resolution (approximately 5x5 km for CR2Met and 2x2 km for RhiresD) that allow us to better represent the meteorology in the complex topography of our study domain. Furthermore, these products have been widely used in hydrological studies in Chile (e.g., Vásquez et al., 2021; Alvarez-Garreton et al., 2021; Araya et al., 2023) and Switzerland

(e.g., Peleg et al., 2020; Fatichi et al., 2015; Tuel et al., 2022). Streamflow records available through the CAMELS datasets were acquired from the national agencies in each country (i.e., the General Directorate of Water of Chile - DGA and the Swiss Federal Office for the Environment - FOEN). Additionally, we retrieve time series of actual evapotranspiration (ET) and soil moisture (SM) from the satellite and reanalysis-based GLEAM dataset (Miralles et al., 2011) aggregated to the catchment scale. We compute topographic characteristics and hypsometric curves - needed to set up the snow routines - using the catchment

outlines from CAMELS and the Multi-Error-Removed Improved-Terrain (MERIT) digital elevation model (Yamazaki et al., 2019).

## 3 Methodological approach

Our methodological approach is illustrated in Figure 2. Four hydrological models are calibrated with data from the CAMELS datasets and streamflow records, using five different formulations of the Kling-Gupta efficiency (KGE) as objective functions.

In addition, we test three streamflow transformations and four different weights applied to the KGE variability term. This calibration exercise results in 60 optimal parameter sets per model and catchment (i.e., 5 KGE x 3 Transformations x 4 Weights). Simulation performance is assessed by comparing observations and model outputs. Specifically, we evaluate model performance based on (1) general goodness-of-fit metrics (Legates and McCabe Jr., 1999; Althoff and Rodrigues, 2021), (2) for extreme events and transitions between them, and (3) for hydrological consistency in different processes related to streamflow,

snow, evapotranspiration, and soil moisture. Event-specific model performance is evaluated at the event scale for droughts, floods, and transitions using categorical indices. In this paper, we use the terms 'formulation' to refer to a specific definition of the KGE (1), 'case' to refer to the application of KGE weights or flow transformations, and 'configuration' to mention the combination of a KGE formulation and a case using certain weights and a specific streamflow transformation. In addition, the cases without weights and/or the linear combination between streamflow without (i.e., Q) and with low-flow transformations

(i.e., 1/Q) will be used as a reference for the comparison of the results. The following sections provide a detailed description of the different methodological steps.

### 3.1 Streamflow extremes characterization

We detect droughts, floods, and drought-to-flood transitions using the method proposed by Götte and Brunner (2024). This method identifies periods of negative streamflow anomalies (droughts) using a daily varying threshold based on a 30 day

rolling percentile of the daily streamflow data, while high streamflow events (floods) are identified using a fixed threshold based on a percentile of the annual maximum streamflow values. We further require that all drought events have a minimum





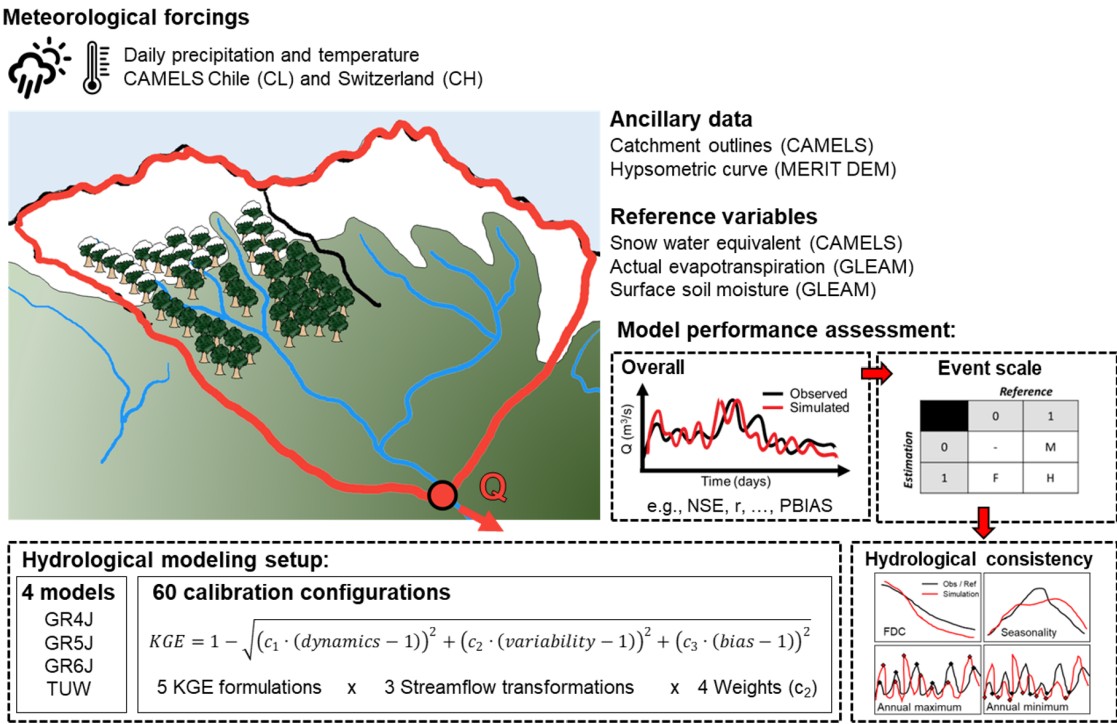

**Figure 2.** Overview of the methodological approach. See text for details

duration of 30 days, and we merge drought events which have a gap of 15 days or less between them (Van Loon and Van Lanen, 2012; Fleig et al., 2006; Tallaksen et al., 1997). This is done to limit the detection of minor events.

Rapid (within 14 days) and seasonal transitions (within 90 days) are defined based on the number of days between the end of the drought to the onset of the flood, as in Götte and Brunner (2024). The thresholds for droughts and floods (associated with the 30th and 40th percentile of the smoothed daily flow and the annual maxima series, respectively) were set to ensure roughly one streamflow extreme event of each type (i.e., drought and flood) per year on average for each catchment. This target was set in order to identify a number of extreme events that results in a statistically representative sample size comparable to the sample size that would be obtained by the commonly used annual maximum approach (e.g., Meylan et al., 2012). To identify those thresholds that met our criteria, we tested different values (see Figure S1 in the Supplementary Material). Figure 3 shows the detection of droughts and floods based on the approach adopted for two recorded transitions in two catchments within the study domain.





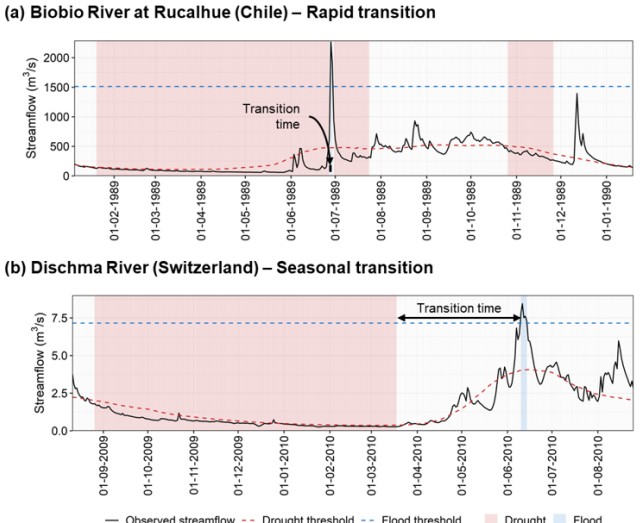

**Figure 3.** Example of the characterization of streamflow extremes and their transitions for two catchments within the study domain. (a) Biobio River at Rucalhue in Chile, and (b) Dischma River in Switzerland.

## 3.2 Modeling approach

### 3.2.1 Hydrological models

We use four conceptual bucket-style rainfall-runoff hydrological models: GR4J (Perrin et al., 2003), GR5J (Le Moine, 2008), GR6J (Pushpalatha et al., 2012), all coupled to the snow accumulation-ablation module CemaNeige (Valéry et al., 2014a, b), and TUWmodel (Parajka et al., 2007), which is based on the HBV model (Bergström and Forsman, 1973). All models have been widely used within the hydrological community during the last decades (Seibert and Bergström, 2022). GR4J, GR5J, and GR6J (with 6, 7, and 8 parameters coupled with CemaNeige, respectively) were chosen to explore how slight changes in

model structure affect simulated streamflow extremes and the TUW model (with 15 parameters) was chosen to explore how more complex models, particularly with respect to the snow routine and the representation of the processes occurring in the production storage, simulate these phenomena. Such simple bucket-type models have the advantage that they enable studying different processes of the hydrological cycle at a lower computational cost than more complex model structures (e.g., Clark et al., 2017; Orth et al., 2015; Poncelet et al., 2017).

The GR4J, GR5J, and GR6J models, which we will refer to collectively as GRXJ for simplicity, share the same genealogy, meaning that they are based on the same core structure. These models can be coupled to the snow module CemaNeige, where the precipitation is partitioned between liquid and solid and simulates snow accumulation and melt (rainfall and snowmelt enter to GRXJ structures). The basic structure of the GRXJ family corresponds to the GR4J model, which includes a parameter for production storage capacity, representing surface processes, and a parameter for routing storage capacity, representing sub-

surface processes. Additionally, GR4J includes an intercatchment exchange parameter (non-conservative function) and a unit





hydrograph parameter that corresponds to the delay between precipitation and streamflow. GR5J adds an additional parameter to the GR4J structure to improve the intercatchment exchange function, while GR6J includes a parameter for exponential storage in parallel to the routing storage in GR4J and GR5J to improve the representation of groundwater processes (i.e., slow runoff).

The TUW model consists of a snow, soil, groundwater (subsurface flow), and a routing routine, similar to the HBV model (Bergström and Forsman, 1973). The snow routine partitions between liquid and solid precipitation and estimates snow accumulation and melt. Rainfall and snowmelt enter the soil routine, where actual evaporation, soil moisture, and recharge are estimated. Then, the recharge flow goes to the groundwater routine, represented by two storages that produce surface runoff and quick flow (upper), and baseflow (lower). The sum of these flows is delayed in the routing routine using a triangular transfer
function. Unlike GRXJ models, which follow a water balance approach to characterize the production storage, TUW estimates evapotranspiration and recharge based on an explicit conceptualization of soil moisture content.

While both CemaNeige and the snow routine implemented in the TUW model follow a degree-day factor approach, there are some differences mainly in i) the characterization of the precipitation phase (TUW allows the existence of a mixed partition between rain and snow), ii) the conditions for snowmelt (free-parameter in TUW model and set as 0°C for CemaNeige), and
iii) the presence (or not) of a parameter to correct for snowfall undercatch (not available in CemaNeige). These differences also explain the number of parameters that each of the snow routines has (two and five for CemaNeige and the snow routine in the TUW model, respectively).

Despite their structural differences and conceptualizations (for further details refer to Astagneau et al., 2021b), these models provide simplified representations of some hydrological states, fluxes, and processes at the catchment scale using precipitation
(P), mean temperature (T), and potential evapotranspiration (PET) at daily time steps as an input. To estimate PET, we use the approach proposed by Oudin et al. (2005) which is based on temperature and includes latitude and day of the year as a proxy for extraterrestrial radiation for its computation. Additionally, as the snow module CemaNeige can be configured in a semi-distributed way discretizing each catchment into elevation bands of equal area based on the hypsometric curve, we considered 10 elevation bands for all evaluated model structures. To make simulations comparable across model structures, precipitation
and temperature for the TUW model were extrapolated following the approach implemented in the GRXJ models through 10 elevation bands, based on the orographic gradients defined by Valéry et al. (2010).

In addition to the parameters associated with each hydrological model, two parameters were included in the calibration process to address issues - such as the catchments placed above the water limit or below the energy limit - pointed out in Figure 1c. Thus, a multiplicative parameter for precipitation and an additive parameter for temperature were included to adjust their
values.

### 3.2.2   Calibration strategy

The parameters of each of the four model structures were calibrated using daily streamflow records and the Shuffled Complex Evolution global optimization algorithm (SCE-UA; Duan et al., 1992) on the period between 2000-2020, defined to capture the current hydroclimatic conditions in the modeling parameters. Different objective functions based on KGE configuration were





used to calibrate each model. In its most general form, the KGE (Eq. (1)) compares simulations to a reference based on three components, i.e. dynamics (e.g., correlation), variability (e.g., standard deviation), and bias (e.g., mean). KGE values range from negative infinite to one, with one corresponding to the optimum. How each component is defined depends on which KGE formulation is used. To the best of our knowledge, there exist five such formulations in the literature (Gupta et al., 2009; Kling et al., 2012; Pool et al., 2018; Tang et al., 2021; Pizarro and Jorquera, 2024, more details in Table 1). Additionally, different scaling factors or weights (i.e., $c_1$, $c_2$, and $c_3$ in Eq. (1)) can be used to give more emphasis to some of the components of the KGE as well as different streamflow transformations to give more weight to specific parts of the flow distribution, (e.g., Thirel et al., 2024; Mizukami et al., 2019). To emphasize low flows, for example, flow can be transformed to the inverse of streamflow (i.e., 1/Q; e.g., Garcia et al., 2017; Wu et al., 2025). Further, linear combinations of the KGE applied to flows without and with transformation (i.e., Q and 1/Q, respectively) have been presented as a useful objective function to find a good compromise between high- and low-flows (e.g., Araya et al., 2023; Knoben et al., 2020; Muñoz-Castro et al., 2023).

$$KGE = 1 - \sqrt{(c_1 \cdot (\text{dynamics} - 1))^2 + (c_2 \cdot (\text{variability} - 1))^2 + (c_3 \cdot (\text{bias} - 1))^2} \tag{1}$$

For each hydrological model and catchment, 60 different objective functions (OF) are implemented based on the possible combinations of the following methodological decisions: (i) 5 KGE formulations (Table 1), (ii) 3 streamflow transformation cases (High, Low, High-Low), and (iii) 4 weights applied to the variability term (i.e., in Eq. (1), $c_2$ = 1, 2, 4, 8). For the low-flow transformation (Low; i.e., using 1/Q), a constant equal to 1% of the mean streamflow is added to the series to avoid zero-flow problems following the recommendations by Pushpalatha et al. (2012). To facilitate the notation associated with the streamflow transformations tested here, we will refer to the case 'Hi' (High) when a certain formulation of KGE is applied to untransformed streamflow (i.e., Q), while 'Lo' (Low) will refer to the case where a low-flow transformation is applied (i.e., 1/Q). To the linear combination of both cases (i.e., 0.5*Hi + 0.5*Lo), we will refer to it as the 'HiLo' case.

### 3.3 Model performance assessment

We evaluated model performance with respect to general performance as described by the Nash-Sutcliffe Efficiency (NSE; Nash and Sutcliffe, 1970) and performance with respect to simulating extreme events as described by the Critical Success Index (CSI). Following a traditional split-sample test approach (Klemeš, 1986; Beven, 2025), the model performance is assessed over two time periods defined as (i) calibration (2000-2020), and (ii) evaluation (1985-1999). The following subsections provide a detailed description of how the model's performance was evaluated.

#### 3.3.1 Overall performance and hydrological consistency

To evaluate how different model configurations simulate different hydrological variables, we compute the Nash-Sutcliffe-Efficiency for different variables including high- and low-flows (i.e., Q and 1/Q), snow water equivalent (SWE), soil moisture (SM), and actual evapotranspiration (ET). We use the NSE to have a metric easily interpreted and recognized by the hydrological community, that allows us to compare results independently across different configurations adopted for model calibration.




**Table 1.** Summary of KGE formulations. In each formulation, the term dynamics stands for the representation of the temporal evolution of the target variable, while the terms variability and bias aim to characterize its distribution.

| KGE formulation | Components | Description | Reference |
|---|---|---|---|
| Original (KGE) | **Dynamics:** Pearson correlation coefficient.<br>**Variability:** Ratio between the standard deviation of the simulated and observed values.<br>**Bias:** Ratio between the mean of the simulated and observed values. | Meta-objective function, oriented to quantify the Euclidean distance between the absolute error associated with each component. Proposed to overcome the problems associated with NSE (e.g., observed mean as baseline, formulation, which could lead to large volume balance errors or favor models/parameters sets that underestimate the observed variability). | Gupta et al. (2009) |
| Modified (KGE_mod1) | **Dynamics:** Pearson correlation coefficient.<br>**Variability:** Ratio between the coefficient of variation of the simulated and observed values.<br>**Bias:** Ratio between the mean of the simulated and observed values. | Modification in the variability component defined in the original formulation (i.e., standard deviation ratio) aimed to ensure that the bias and variability ratios are not cross-correlated. | Kling et al. (2012) |
| Non-parametric (KGE_np) | **Dynamics:** Spearman's rank correlation coefficient.<br>**Variability:** Error between all ranked simulated and observed values (i.e., flow duration curve) normalized to remove the volume information and keep only the distribution signal.<br>**Bias:** Ratio between the mean of the simulated and observed values. | Reformulation of the variability and correlation terms in a non-parametric way to address the implicit assumptions of linearity and normality of the data in the original formulation. | Pool et al. (2018) |
| Modified v2 (KGE_mod2) | **Dynamics:** Pearson correlation coefficient.<br>**Variability:** Ratio between the standard deviation of the simulated and observed values.<br>**Bias:** Ratio between the mean of the simulated minus the observed values and the standard deviation of the observed values. | Modification in the bias component defined in the original formulation aimed to avoid anomalously negative values when the mean value is close to zero. | Tang et al. (2021) |
| K-Moments (KGE_km) | **Dynamics:** Pearson correlation coefficient.<br>**Variability:** Ratio between the coefficient of variation of the simulated and observed values defined from unbiased estimators of non-central K-moments (alternative formulation for the second moment).<br>**Bias:** Ratio between the mean of the simulated and observed values. | Modification in the variability component defined in the original formulation oriented to make it less sensitive to outliers and non-normal distributions. | Pizarro and Jorquera (2024) |

To minimize the influence of biases in ET and SM estimates on the estimates of model performance - due, for example, to catchment-scale averaging (e.g., Rouholahnejad Freund et al., 2020) or uncertainties related to the GLEAM algorithm (e.g., Jahromi et al., 2022) - we only consider the signal (i.e., timing and variability) associated with these variables rather than their absolute values.

### 3.3.2 Detection of streamflow extreme events

To assess model performance with respect to the detection of streamflow extremes and their transitions, we use the critical success index (CSI; Eq. (2)), which is formulated based on the number of hits (H; events identified both in the reference/observation and the simulation), misses (M; events only identified in the reference/observation), and false alarm events (F; events identified only in the simulation). The CSI provides a measure of the overall performance of the simulations in terms of detecting observed events and its values vary between zero and one, with one being the optimum. We define hits as simulated events overlapping at least 50% with their observed counterparts. Additionally, for the detection analysis, a tolerance window of 30 and 5 days is defined before the onset and after the end of an observed drought and flood event, respectively. This window allows for considering the differences of gridded meteorological products with reality and how these can affect the timing of the simulated events (i.e., early or late compared to the observed streamflows).



CSI = H/(H + M + F)                                                                                     (2)

## 3.4    Assessment of the relative importance of modeling decisions

To assess the relative importance of each modeling decision on the detection of streamflow extremes and their transitions, we conduct an analysis of variance (ANOVA; Fisher, 1992; Kaufmann and Schering, 2014). The ANOVA allows us to study the relationship between different modeling decisions (e.g. choice of structure and different decisions related to calibration) and

quantify their relative importance in explaining the total variance in the target variable (e.g., CSI). Thus, by dividing the total variance into different groups, genuine sources of variation that are not explained by chance can be identified. We assume that the total variance (TV) in the target variable could be mainly explained by the differences between catchments (CT), hydrological models (HM), KGE formulations (KGEf), streamflow transformations (QTR), and KGE component weights (W). If, for example, weights do not have a significant impact on the detection of streamflow extremes, we would expect a low

value for the term "W", that is a lower relative importance (i.e., W/TV) for explaining the total variance with respect to other decisions. Based on this conceptualization and considering a residual term (RS) that groups all the interactions between decisions and the variance that we cannot explain from them, the ANOVA can be expressed as follows:

TV = CT + HM + KGEf + QTR + W + RS                                                                            (3)

## 3.5    Identification of important processes in simulating drought-to-flood transitions

To identify the most important processes in simulating drought-to-flood transitions, we ask what explains the accurate detection of events. To address this question, we analyze the relative importance of each model parameter in estimating the CSI through an ANOVA test applied per catchment. This analysis considers the 60 alternative configurations (i.e., parameter sets) available per model and catchment and uses the total variance explained by each parameter as a proxy for the importance of the associated variable/process. In addition, we explore which components of the modeling chain could be improved to better

capture transitions. For this, we perform a similar ANOVA analysis, using the bias associated with different hydrological signatures instead of model parameters to estimate their relative importance in explaining the total variance of the CSI. Finally, to identify potential controls on the relative importance of some parameters in our study domain, we compute the Spearman's rank correlation coefficient between the most important parameters and a selection of physiographic and hydroclimatic catchment descriptors. The approach used to analyze the relative importance of the parameters explaining the variance of the CSI

may have problems if the parameters do not show enough variation between the different configurations. However, despite the similarities in the configurations used for calibration, almost all the parameters show high variability among the calibrated parameter sets per catchment (see Figure S14 in Supplementary Material).




## 4 Results

### 4.1 Suitability of KGE for calibrating models aimed at simulating drought-to-flood transitions

Model performance described by general performance metrics such as KGE during calibration and evaluation is often used as a proxy for how well a model represents streamflow properties such as extreme events. To explore how valid this assumption is, i.e. how closely general model performance is linked to its capability of representing extreme events, we compare the performance of the model during the calibration and evaluation period based on the objective function defined for calibration - the original KGE formulation configured with unweighted HiLo (i.e., $c_2 = 1$ and HiLo = 0.5*KGE(Q) + 0.5*KGE(1/Q)) -

with the performance in detecting droughts, floods and their transitions based on the critical success index (CSI; Figure 4). Our comparison clearly shows that while overall performance described by the KGE can potentially be a useful proxy for a model's performance in capturing droughts (e.g., CSI ranges from 0.23 to 0.79 for GR4J and from 0.21 to 0.74 for TUW), it is not for floods and transitions. A high KGE does not necessarily imply a high CSI for these two types of events. Model performance varies across catchments and different model structures both for KGE and CSI (Figure 4). For droughts and transitions (panels

a.1-b.1 and a.3-b.3, respectively), GR4J and TUW show comparable results both in terms of the median KGE (e.g., 0.85 and 0.80 in evaluation) and CSI (e.g., 0.59 and 0.51 for droughts) as well as their variability (error bars), but the model performance decreases when GR5J and GR6J are analyzed. For floods (i.e., panels a.2 and b.2), the GR4J outperforms the other models tested both in terms of general performance (median KGE is 0.06-0.11 and 0.04-0.18 larger than in other models for calibration and evaluation period respectively) and the detection of extreme events (median CSI is 0.03-0.13 larger than in other models

depending on the type of event).

While KGE is not necessarily a good proxy for how well a model captures extreme events (especially floods and transitions), some specific KGE formulations might be better suited for this task than others. Next, we therefore evaluate to what extent different adjustments in the 'basic' configuration used for the analyses presented above can (or cannot) improve the performance in detecting streamflow extreme events and, particularly, drought-to-flood transitions.

### 4.2 Impacts of KGE configurations on drought-to-flood transition simulations

#### 4.2.1 KGE weights

To assess the added value of applying weights to the variability term of the KGE for detecting independent extreme events and their transitions in terms of the critical success index (CSI), we compute the difference between the non-weighted case (reference) and the application of three different weights (alternative cases) for each KGE formulation and streamflow transformation

(Figure 5a for the GR4J model). This comparison shows that, in general, the use of weights does not consistently improve the ability of the models to detect streamflow extreme events or transitions between them (median difference is centered around 0). While there are some catchments where the introduction of weights improves model performance (catchments with negative differences), there are as many catchments where the introduction of weights decreases performance (catchments with positive differences). Figure 5b provides the same comparison but uses as a reference an unweighted HiLo configuration, showing sim-





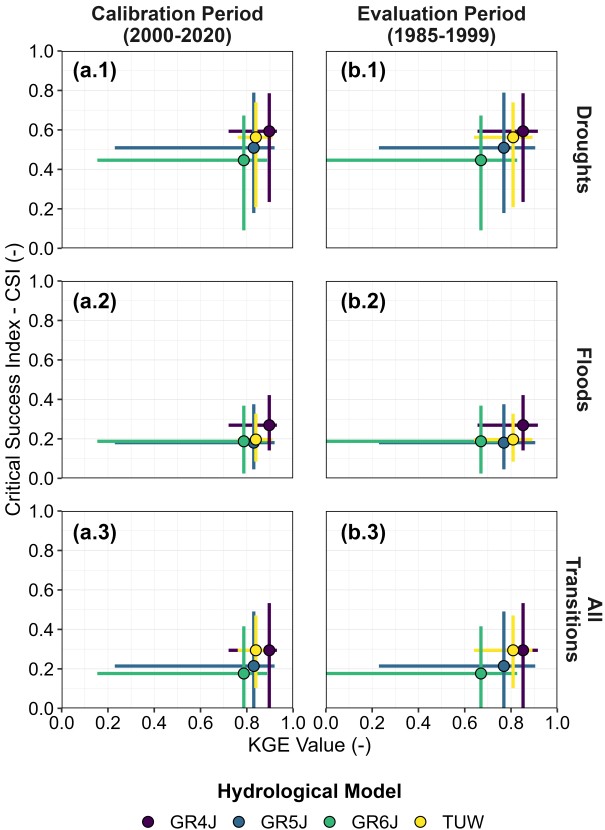

**Figure 4.** Comparison between the Kling-Gupta Efficiency (KGE) for the (a) calibration and (b) evaluation period and the Critical Success Index (CSI) for (1) droughts, (2) floods, and (3) transitions, based on the simulations with the models GR4J, GR5J, GR6J, and TUW calibrated with the unweighted HiLo original KGE formulation as the objective function. CSI is calculated for the whole period (i.e., equal values for both the calibration and evaluation). The error bars are associated with the 10th and 90th percentiles across catchments, while the central shape is associated with the 50th percentile.

ilar results (i.e., no consistent improvements and median centered around zero). However, there are some differences between different streamflow transformations and weights in the representation of floods (panel 2 in Figure 5b), with the unweighted HiLo being better in approximately 75% of cases compared to the weighted low-flow transformation. This finding generalizes to the other three model structures (see Figures S3 and S4 in the Supplementary Material).

These results highlight that, in the context of a large-sample study, the use of weights for the variability term of the KGE
is not beneficial as it does not consistently improve the model's performance in detecting streamflow extremes and their transitions, and its usefulness will depend on the characteristics of the study domain (see Figures S5 in the Supplementary Material). Additionally, our results show that, for the task of modelling drought to flood transitions, an objective function which puts equal weight on high- and low flows (i.e. HiLo) improves model performance as floods are much better captured in




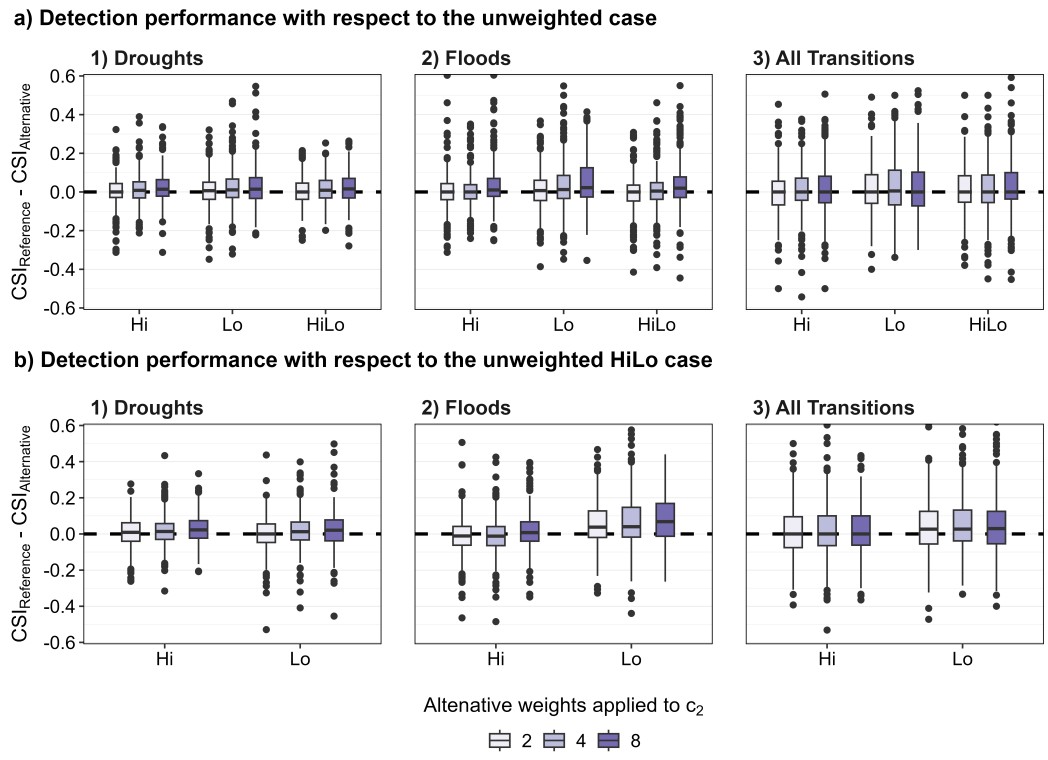

**Figure 5.** Difference in the CSI for GR4J simulations using model calibrations with no weights (reference) versus different weights (alternative) for (1) droughts, (2) floods, and (3) transitions. In (a), each alternative is compared with its unweighted analog, while in (b) the reference includes the HiLo transformation. Values above (below) 0 indicate better (worse) performance of the reference compared to the alternative.

this configuration than in configurations which rely on low-flow transformations (i.e., Lo) and equally well represented as in the configuration without transformation (i.e., Hi).

### 4.2.2 Streamflow transformation

Considering the HiLo (i.e., 0.5*KGE(Q) + 0.5*KGE(1/Q)) configuration as a reference (see Figures S6-S8 in the Suplementary Material for more details about its overall suitability), we assess the role of streamflow transformation on model performance for simulating high and low flows, snow water equivalent, actual evapotranspiration, and soil moisture (Figure 6a for the GR4J model as an example). We find that a transformation which puts equal weight on high- and low flows has the best performance on average, independently of the KGE formulation adopted for the calibration (different colors) and independent of the variable used for the evaluation (different columns). In particular, the results for high- and low flows (i.e., NSE(Q) and NSE(1/Q), respectively) highlight the added value of HiLo compared to transformations focusing on high and low flows individually. Between 50-90% of catchments profit from the HiLo transformation depending on the model and objective function used for calibration. HiLo improves the performance in simulating low (high) flows in comparison to using only high (low) streamflow



transformations during model calibration without compromising the model's ability to adequately represent the opposite flow type (median difference is centered around 0 for NSE(Q)-Hi and NSE(1/Q)-Lo). For the other variables, besides better performance on average, the added value of HiLo compared to the transformations focusing on high and low flows only is less clear. Additionally, there exists substantial variability among catchments. These findings again generalize to the other model structures tested (Figures S9a and S10a in Supplementary material).

### 4.2.3 KGE formulation

Figure 6b compares - for the GR4J model as an example - the unweighted HiLo case for all the different KGE formulations against the original KGE formulation (i.e., Gupta et al., 2009) as a benchmark. We find that using a modified KGE formulation, rather than the original, does not substantially improve model performance for either of the variables evaluated (i.e., Q, 1/Q, SWE, ET, and SM). However, there also exists substantial variability across catchments. Some catchments show an improvement in model performance when using alternative KGE formulations while others show a decrease in model performance. A small difference in the performance achieved through disparate formulations may be found in the high-flow representation (i.e., NSE(Q)), for which the original KGE provides slightly better values on average than the alternatives. Similarly, for SWE, ET, and SM, the added value of the original KGE formulation compared to the alternatives is less clear. In short, the differences in NSE between model formulations are close to zero, and the changes which do occur are not systematic or consistent. This suggests that the choice of the KGE formulation does not play a dominant role in the overall performance of the model over the study domain. These findings are consistent across other model structures tested (Figures S9b and S10b in Supplementary material).

## 4.3 Importance of model structure

Our results show that the detection of low-flow anomalies is typically more reliable than high-flow and transition detection (Figure 7). However, there are substantial differences in the detection rate depending on the hydrological model structure used for simulating extreme events. GR4J overall has the best model performance independently of the KGE formulation chosen for calibration (median CSI values across KGE formulations around 0.58, 0.26, and 0.31 for droughts, floods, and transitions, respectively). The performance of the GR5J and GR6J models decreases compared to GR4J (changes between 0.05 and 0.13 depending on the model and type of extreme event). This suggests that increasing model complexity decreases rather than increases model performance. However, these decreases in performance cannot be directly attributed to increases in the complexity of the model because the TUW model shows comparable results to GR4J despite its more complex structure (median CSI values across KGE formulations around 0.56, 0.19, and 0.28 for droughts, floods, and transitions, respectively). If model structures are compared for Switzerland and Chile, the same conclusions can be drawn in terms of model performance, with comparable results between GR4J and TUW and lower performance of the GR5J and GR6J. However, the detection of extreme events is more challenging in catchments located in Chile compared to those located in Switzerland, with differences in the median CSI between countries being around 0.23, 0.02, and 0.13 for droughts, floods, and drought-to-flood transitions,





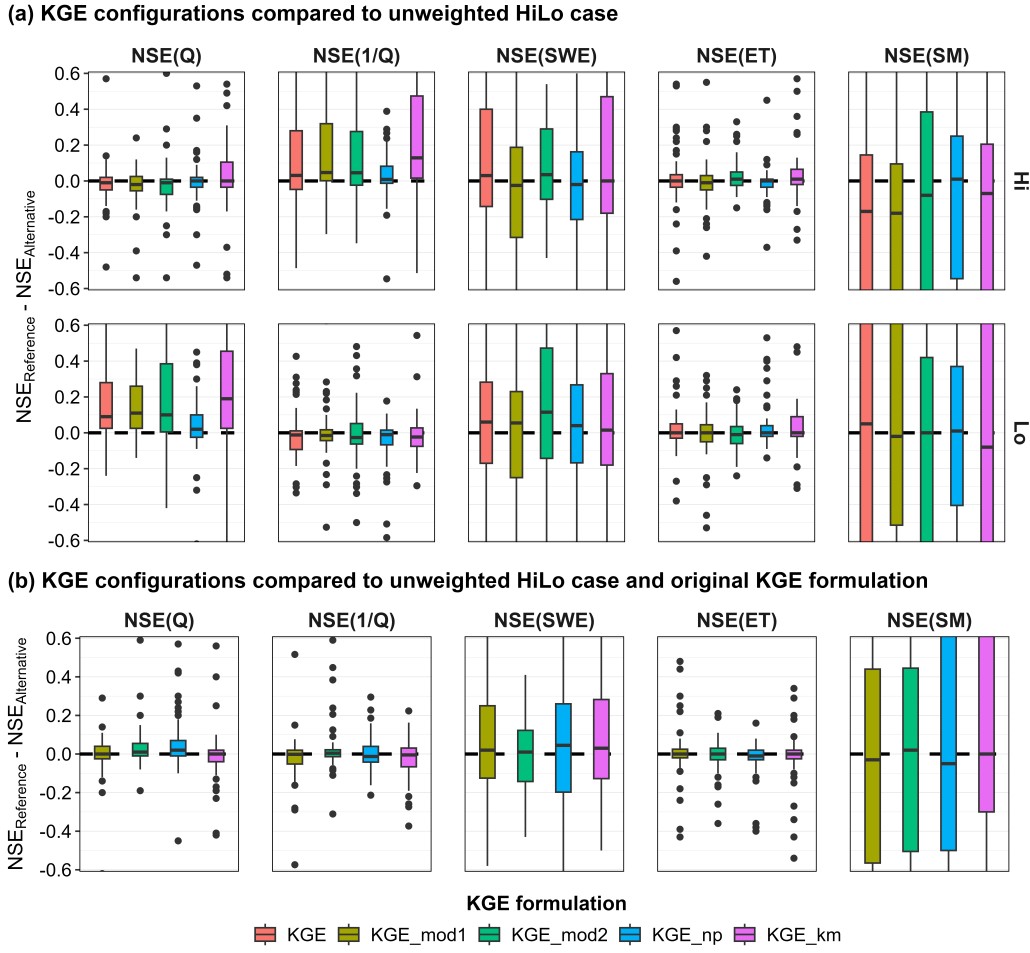

**Figure 6.** Difference in the NSE for GR4J simulations of different hydrological variables in the evaluation period obtained with different KGE configurations. In (a), each alternative configuration is compared to its unweighted HiLo analog, while in (b) the reference represents the original KGE formulation. Values higher (lower) than 0 represent a better (worse) performance of the reference compared to the alternative in simulating high-flows (NSE(Q), first column), low-flows (NSE(1/Q), second column), snow water equivalent (NSE(SWE), third column), actual evapotranspiration (NSE(ET), fourth column), and surface soil moisture (NSE(SM), fifth column).

respectively (see Figure S11 in Supplementary Material). In summary, the GR4J and TUW models seem to be the models best suited for simulating droughts and floods among the models considered in this study.

Different model structures can result in similar streamflow simulations even though they represent hydrological fluxes and states in different ways. To illustrate this, we compare simulated fluxes obtained for an observed seasonal drought-to-flood transition in the Dischma river in Switzerland across the four hydrological models (Figure 8). While three out of four models capture the transition event successfully (GR6J fails in capturing its timing) and show similar temporal patterns of ET,



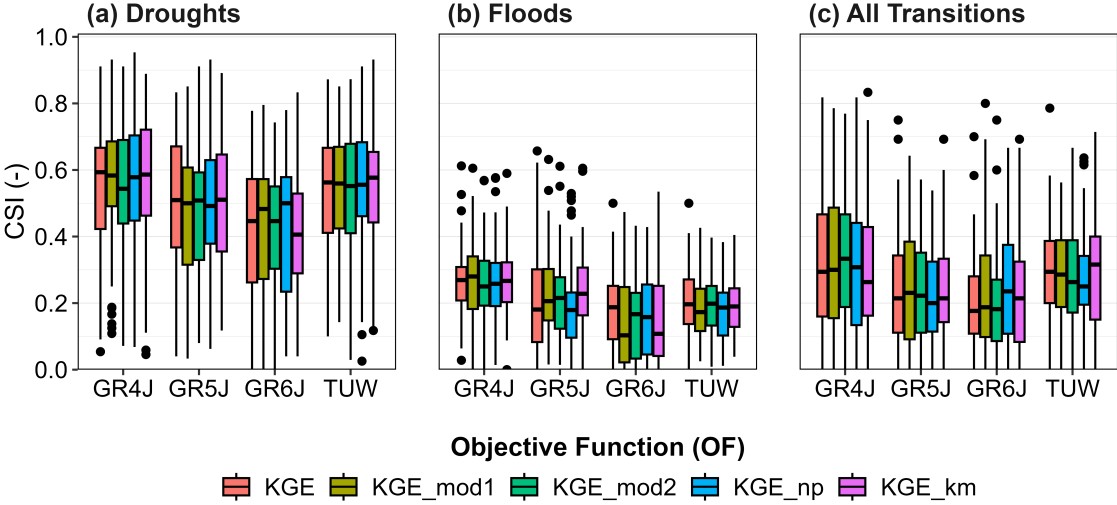

**Figure 7.** Critical Success Index (CSI) for (a) droughts, (b) floods, and (c) drought-to-flood transitions, based on the simulations with GR4J, GR5J, GR6J, and TUW calibrated with different unweighted HiLo KGE formulations as objective functions (different colors).

snowmelt, and SWE, the contribution of baseflow (presented as a percentage of total runoff) and soil moisture varies strongly
among them. Consequently, the analysis of the drivers associated with such transition events will vary depending on which
model structure is analyzed. Although there is a high agreement between the models for the detection of the event in this
example case (i.e., 3 out of 4), this is not necessarily the case for all the events and catchments catchment (Figure 4).

## 4.4    Relative importance of different modeling decisions

Consistent with our earlier findings (Figure 7), the results of the ANOVA analysis show that the most important modeling
decision is the choice of a suitable model structure, followed by the choice of the streamflow transformation, and the differences
between catchments. In contrast, the choices of KGE formulation and weights do not have a strong impact on the performance
of simulating streamflow extremes. The relative importance of the methodological choices is similar when analyzing other
categorical indices such as probability of detection (POD), false alarm ratio (FAR), and frequency of bias (fbias), as well
as the model performance in simulating high and low flows, SWE, ET, and SM, as evaluated by NSE (see Figure S12 in
Supplementary Material). For rapid transitions, the difference between catchments is more important for explaining the CSI
values than it is for seasonal transitions. This difference indicates that the detection of rapid transition events may depend on
the catchment attributes (e.g., mean elevation, streamflow regime, etc.).





**Figure 8.** Example of how different hydrological fluxes and states are simulated for an observed drought-to-flood transition in the Dischma river (Switzerland) with the GR4J, GR5J, GR6J, and TUW hydrological models calibrated with the unweighted HiLo original KGE formulation.




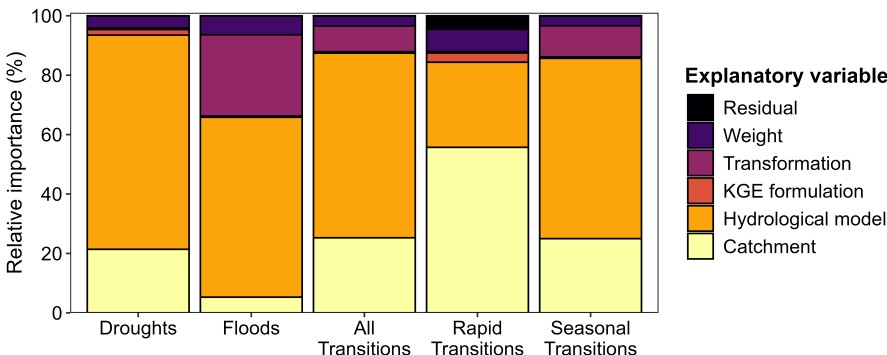

**Figure 9.** Results of the analysis of variance (ANOVA) applied to the Critical Success Index (CSI) for droughts, floods, all drought-flood transitions (i.e., rapid and seasonal), rapid transitions, and seasonal transitions.

## 4.5 Model performance depends on catchment characteristics

Model performance in simulating extreme hydrological events and their transitions depends on catchment characteristics (Fig-
ure 10). To identify catchments where model performance is better compared to others, we explore the relationship between model performance and catchment characteristics using Spearman's rank correlation coefficient. For this, we focus on the CSI obtained for the different types of extreme events of interest (droughts, floods, and transitions) generated with the GR4J and TUW models calibrated with the unweighted Hilo original KGE formulation (Figure 10). Our results show that drought-to-flood transitions are more difficult to capture in semi-arid (negative correlation between aridity index and CSI), high-mountain
(negative correlation between mean elevation and CSI), and flashy (negative correlation between the slope of the flow duration curve and CSI) catchments than in humid low-elevation catchments with high streamflow elasticity to precipitation. This result is generalizable to the other models and different KGE formulations tested (see Figure S13 in the Supplementary Material).

## 4.6 Linking model performance to hydrological processes during streamflow extremes

We perform an ANOVA test to analyze the relative importance of different model parameters for the detection of streamflow
extremes and transitions between them (Figure 11; for the extended version see Figure S15 in the Supplementary Material). Here, we show that some model parameters are relatively more important than others (e.g., X4 for floods in GRXJ models), but that the relative importance of a given parameter can vary substantially across catchments. All of the tested hydrological models show a high importance of the parameters aimed to adjust the forcings (i.e., dP and dT for all the models as well as SCF in TUW model, which seeks to correct for the snow undercatch), highlighting the need for adequate forcing to improve the estimation
of extreme hydrological events. For the GRXJ models, parameters X3 (routing store capacity) and X4 (unit hydrograph time constant) are more important in the simulation of low and high flow compared to the rest of the parameters, which is accentuated even more when the base structure is made more complex (i.e., GR6J). In the TUW model, which has more parameters than the GRXJ structures, the relative importance of each of the parameters is more equal and their relative importance is low, except for




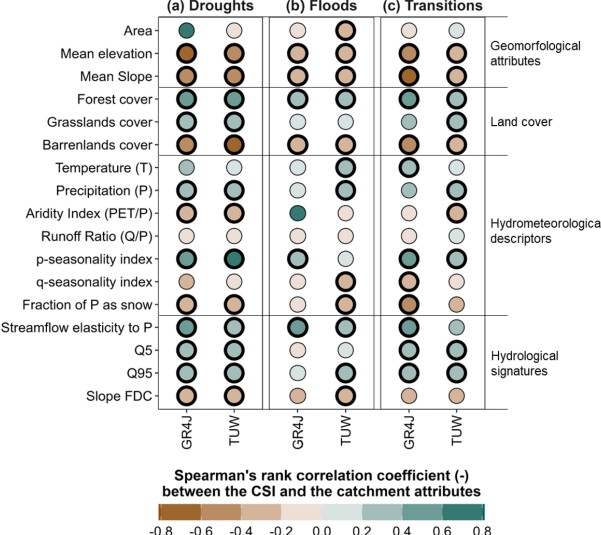

**Figure 10.** Spearman's rank correlation coefficient between different catchment attributes and the CSI for (a) droughts, (b) floods, and (c) drought-to-flood transitions, based on the simulations with GR4J and TUW calibrated using the unweighted HiLo original KGE formulation as the objective function. The circles with thick outlines indicate statistically significant correlation coefficients at a the 5% significance level.

the parameter k0 (storage coefficient for very fast response) during floods. Additionally, we compare the relative importance of biases in hydrological signatures in the detection of streamflow extremes and their transitions to identify key processes to be captured to better simulate them, finding that model performance is mainly driven by an adequate representation of streamflow timing (Figure S16 in the Supplementary Material).

The relative importance of the parameters changes depending on the type of streamflow extreme (Figure 11). We select the most important parameters in each model to explore their relationship with some physiographic and hydroclimatic catchment descriptors (see Figure S17 in the Supplementary Material). For the GRXJ models we considered CN2 (degree-day melt coefficient), X3 (routing store capacity), and X4 (unit hydrograph time constant), and for the TUW model DDF (degree-day factor), k0 (storage coefficient for very fast response), and bmax (maximum base of the triangular transfer function). Our results show that while topographic characteristics (i.e., area, mean elevation, slope) show more correlation with the relative importance of snow and river routing parameters for floods, runoff routing parameters become less important during rapid transitions depending on land cover and seasonality. In addition, snow and runoff routing parameters play a key role during seasonal transitions.



**Figure 11.** Relative importance of parameters for explaining the Critical Success Index (CSI) for models (a) GR4J, (b) GR5J, (c) GR6J, and (d) TUW based on the results of an analysis of variance (ANOVA).

## 5 Discussion

### 5.1 Simulation of compounding streamflow extreme events

We find that the hydrological models tested are better at detecting droughts (median CSI across catchments and KGE formulations: 0.45-0.58 depending on the model) than floods (median CSI across catchments and KGE formulations: 0.15-0.26 depending on the model), and their performance in detecting drought-to-flood transitions is closely related to the performance in detecting floods (Figure 4). This difference in drought and flood simulation performance can be attributed to the different timescales during which these two types of extreme events develop: while droughts vary in duration from months to years (or decades), floods develop, and may also subside, in a matter of days. This is consistent with the poor performance of all



the models tested in capturing rapid transitions (median CSI equal to zero when rapid and seasonal transitions are analyzed separately; not shown). Moreover, only in thirteen basins - based on different configurations - we obtained CSI values greater than zero for rapid transitions (Table S2 in Supplementary Material). Our analysis highlights that these fast processes are rather difficult to capture in conceptual models, even when they have been calibrated.

## 5.2 Good general performance does not imply that extremes are well detected

Our results highlight that a good general model performance as e.g. described by the KGE does not necessarily imply a good performance in detecting streamflow extremes. Even models with high KGE values ($> 0.6$) struggle to capture extreme events, in particular floods and transitions from droughts to floods (Figure 4). These findings are aligned with previous studies which discussed the potential of KGE to represent high-flow values or capture flashy dynamics (e.g., Astagneau et al., 2022; Brunner et al., 2021; Mathevet et al., 2020). For instance, Astagneau et al. (2021a) demonstrated that the relationship between KGE

values and model performance in terms of simulating summer floods is weak. Similarly, Bruno et al. (2024) have shown that during extreme low-flows model performance is usually lower than during normal flow conditions. Spieler and Schütze (2024) showed that by focusing on basic catchment-scale behaviors, the KGE lacks the capacity to provide information about detailed processes, leading to gaps between model accuracy (i.e., how well a model matches simulations with observations) and adequacy (i.e., how well a model captures key processes and behaviors of the observed system). These findings suggest that

the traditional evaluation of hydrological models through goodness-of-fit metrics such as KGE or NSE must be accompanied by an explicit examination of their capability to simulate and detect streamflow extreme events, e.g. by using metrics such as the CSI.

## 5.3 The importance of different modeling decisions for simulating streamflow extremes and their transitions

Our results show that model structure is the most important modeling decision for capturing extreme events and their transitions
(Figure 9). Among the structures tested, the GR4J model is the one that shows the best performance both for the simulation of independent extreme events and for transitions (Figure 7). The TUW model shows similar results but with a lower flood detection performance (median CSI between KGE formulations of 0.19 compared to 0.26 for GR4J). These deficiencies in flood simulation performance translate to deficiencies in capturing drought-to-flood transitions (Figure 4 and Figure 7). The lack of an explicit structural component that allows for the simulation of floods that occur under dry conditions with low soil

moisture could explain the poor performance associated with this type of compound event. Astagneau et al. (2022) highlighted that conditioning the storages and fluxes of a lumped conceptual hourly-timestep model on rainfall intensities could benefit model performance in catchments with a fast response to precipitation (i.e., flashy-catchments). For droughts, van Kempen et al. (2021) have shown that the magnitudes of the low-flow events are significantly affected by alterations in the architecture of the upper and lower storages, which is consistent with the changes in performance among the GRXJ models, where small

structural modifications lead to important changes in the detection of these events. van Kempen et al. (2021) have shown that multiple hydrological processes significantly affect the magnitude of high- and low-flow events, which implies that the



model structure is an important source of uncertainty. This supports our findings, which show that in the context of studying streamflow extreme events, the choice of hydrological model is the most important decision (Figure 9).

We demonstrated that the capability to identify streamflow extreme events and their transitions in simulations is model-
dependent. However, the change in performance in detecting extreme events does not necessarily depend on the number of parameters or model complexity. We obtained similar performances between GR4J and TUW despite their structural differences (Figure 7), unlike GR5J and GR6J whose performance declines in comparison with GR4J. Thus, how hydrological models represent the interactions, flows and states in the hydrological cycle (i.e., model parametrization) could be influencing their performance (e.g., Knoben et al., 2020; Beven and Chappell, 2021; van Kempen et al., 2021). Several studies have
highlighted that including a more detailed representation of hydrological processes in models does not necessarily imply better performance (e.g., Orth et al., 2015; Valéry et al., 2014a). This is because more detailed representations require more information to characterize the system of interest, which is not necessarily always available at high quality (e.g., land cover maps, distributed forcings, high-resolution digital terrain model, soil properties, etc.). Recently, Santos et al. (2025) found that models with varying complexity can lead to similar robustness issues. Consequently, beyond the complexity of the selected modeling
structure, understanding the extent to which these are useful for studying the phenomenon of interest is key, suggesting the need to improve strategies for identifying the suitability of a given structure (e.g., Spieler and Schütze, 2024; Knoben et al., 2020).

Compared to the choice of model structure, the choice of objective function is less important (Figure 9). However, model performance can be optimized both in terms of general performance (NSE) and the representation of extreme events (CSI)
under (a) the application of equal weights to all components of the KGE (Figure 5) and (b) the application of a streamflow transformation that focuses on both high and low flows (Figure 6a). Our comparison also highlights that the potential benefit from adjusting these choices (e.g. using other weights or other transformations) varies widely between catchments (Figure 5). This is in line with the findings of Mizukami et al. (2019), who found that the influence of weights on model performance depends on model structure and catchments characteristics. While none of the tested modifications in the objective function
consistently improves the simulation of streamflow extremes across all catchments in the study domain, some of the alternative KGE formulations could improve the simulation of certain variables. For example, KGE_np (Pool et al., 2018) shows median differences close to zero when both NSE(Q) and NSE(1/Q) are assessed independently of the flow transformation applied. Its performance highlights the ability of this formulation to capture the distribution of flows, which can be directly attributed to its non-parametric conceptualization.

The meteorological forcing used to simulate streamflow extreme events can also have a major impact on the model's performance (Figure 11). Our results show a high importance of the parameters aimed to adjust the forcings (i.e., dP and dT for all the models as well as SCF in TUW model, which seeks to correct the snow undercatch) for detecting streamflow extremes and their transitions. Obtaining reliable meteorological forcing has been highlighted as one of the key challenges in hydrological modeling studies (e.g., Brunner, 2023; Döll et al., 2016). Several studies have shown that errors in meteorological forcing data
have a large influence on the simulation of snow processes (e.g., Tang et al., 2023; Günther et al., 2019) because of their influence on the precipitation phase (Harder and Pomeroy, 2014), or significant impacts on the partitioning between evaporation and



runoff (e.g., Nasonova et al., 2011). These previous studies and our findings suggest that an improvement in the spatiotemporal representation of precipitation and temperature, as well as of the potential interactions between these variables could contribute to improving the representation of compounding streamflow extreme events in hydrological models.

## 5.4 Limitations and recommendations for future work

Our model and methods intercomparison focused on the simulation of extreme streamflow events, which required the choice of specific event definitions. Here, we have defined hydrological droughts and floods using threshold based-approaches. The thresholds have been adjusted in a way to identify, on average, one event per year and catchment. This methodological choice does to a certain degree influence the outcomes of the comparison. To describe this influence, we tested different thresholds for defining streamflow extreme events. The results of this sensitivity analysis show that the use of more relaxed thresholds to define droughts (i.e., higher percentiles) can improve the performance of the models in detecting such events. However, we did not find such an effect for floods and transitions, for which we find similar model performances independently of the thresholds used (see Figure S2 in Supplementary Material). The improvement in drought detection when the threshold is relaxed can be explained by the observation that models generally struggle more with more extreme events (e.g., Bruno et al., 2024), which are relatively less numerous if the threshold is lowered. While our study shows that threshold choice does not substantially affect model performance in terms of transition events, the method used to define streamflow extreme events can have a major impact on the characteristics of the transition events identified.

To support our analysis, we tested four bucket-type hydrological models widely used within the hydrological modeling community (Addor and Melsen, 2019). Even though these models are at the lower end in terms of model complexity (Hrachowitz and Clark, 2017), they allowed us to perform a comprehensive analysis of different model structures at a lower computational cost than when using models with more complex structures (e.g., Clark et al., 2017; Orth et al., 2015; Poncelet et al., 2017). Furthermore, previous studies and our study have shown that more complexity does not necessarily imply better performance (Figure 7; e.g., Li et al., 2015; Merz et al., 2022).

Our results provide some insights on which avenues of future research could benefit drought-to-flood transitions modeling: exploring the use of modular platforms and a multi-model ensemble approach to quantify model uncertainty and identify more suitable model structures (e.g., Saavedra et al., 2022); improving our understanding of the role of the spatial variability of precipitation for accurate flood simulations (e.g., Macdonald et al., 2025; Astagneau et al., 2022); assessing the benefits of model runs at a subdaily timestep (e.g., hourly); and exploring alternative data-driven modeling approaches such as long short-term memory (LSTM) networks (e.g., Frame et al., 2022; Anshuka et al., 2022). Additionally, exploring relationships between the occurrence or characteristics (e.g., duration, severity) of this type of hydrological extreme events and some large-scale climate patterns (Garreaud et al., 2020; Marengo and Espinoza, 2016; Sun et al., 2016; De Luca et al., 2020) could improve their predictability.



# 6    Conclusions

We performed a methods and models intercomparison study to (i) explore to what extent hydrological models can simulate
drought-to-flood transition and (ii) identify suitable modeling choices aimed at capturing these compound extreme events.
For this intercomparison, we calibrated four conceptual bucket-type hydrological models (GR4J, GR5J, GR6J, and TUW) for
63 catchments in Chile and Switzerland using 60 different configurations of the Kling-Gupta Efficiency (KGE) as objective
functions, based on five KGE formulations, four scaling factors, and three streamflow transformations. Based on the results of
this intercomparison, we draw the following conclusions:

1. Drought events are better captured by hydrological models than flood events. As a consequence, a model's capability
   in detecting transitions from drought-to-flood events mainly depends on how well it captures flood events. In addition,
   model performance with respect to capturing all three event types varies widely between catchments.

2. A satisfactory general model performance as expressed by the KGE or NSE does not guarantee a good performance
   in terms of detecting streamflow extremes and their transitions. While KGE can serve as a rough proxy for low-flow
   performance, it cannot for high-flows and drought-to-flood transitions. Consequently, assessments of the suitability of
   hydrological models for simulating extreme events and their transitions should be complemented with metrics describing
   extreme event detection performance such as the critical success index (CSI).

3. The most important modeling decision when it comes to simulating floods, droughts, and their transitions is the choice
   of a suitable model structure. Here, we demonstrate that the GR4J and TUW models have similar performance - with
   GR4J being slightly better at detecting floods and transitions - and adding model complexity by increasing the number
   of parameters does not necessarily improve the representation of extreme events.

4. In contrast, the choice of the objective function and its exact configuration are less important. The choice of a suitable
   streamflow transformation can improve the simulation of extreme events to a certain degree. Specifically, a joint focus
   on high and low flows by equally weighting the two streamflow transformations in the objective function (referred to as
   HiLo in our analysis) can improve model performance without compromising its ability to capture streamflow extremes.
   However, the choice of the exact KGE formulation and the use of weights for the variability term of the KGE do not
   substantially affect the simulation of extreme events and the direction of this effect depends on the catchment.

5. The model parameters targeted at adjusting the meteorological forcing are relatively more important than other parame-
   ters in explaining the CSI variations. This highlights the need for adequate meteorological forcing data that adequately
   represent the spatiotemporal properties of precipitation and temperature.

6. A model's performance in simulating streamflow extremes and transitions primarily depends on how well it captures
   streamflow timing rather than other hydrological signatures or variables such as evapotranspiration or snow water equiv-
   alent.



7. Drought-to-flood transitions are more difficult to capture in semi-arid, high-mountain, and flashy catchments than in humid low-elevation catchments.

This methods and model intercomparison highlights that simulating streamflow extremes and their transitions is not a trivial modeling task and continued research is needed to improve model performance. The results of this intercomparison study suggest that time is best invested when focusing on improving model structures rather than calibration procedures. Specifically, hydrological model development should focus on improving the representation of processes and components associated with the temporal dynamics of discharge, such as routing or the soil response to intense snowmelt and rainfall. In addition, the strong link between model performance and parameters aimed at correcting precipitation inputs suggests that the representation of extreme events could also be improved by investing in the quality of meteorological forcing datasets. Investments in improving the simulation of extreme events and their transitions are crucial because hydrological models can support process understanding related to compounding streamflow extremes, be used to forecast such events at short time scales, and to project future changes in the occurrences of drought-to-flood transitions – applications that are strongly needed to ensure society's preparedness for these impactful events.

*Code availability.* The code used in this study is available from the corresponding author upon reasonable request.

*Data availability.* The data used to produce the results shown in this paper - such as parameter sets used to generate the simulations and performance metrics - are publicly available through Zenodo (Muñoz-Castro et al., 2025, https://doi.org/10.5281/zenodo.14803501). CAMELS-CL (Alvarez-Garreton et al., 2018a) is available at PANGAEA (Alvarez-Garreton et al., 2018b) and https://camels.cr2.cl/ while CAMELS-CH (Höge et al., 2023a) at Zenodo (Höge et al., 2023b). The GLEAM3.8a dataset (Miralles et al., 2011) is available upon request at https://www.gleam.eu/.

*Author contributions.* EMC, BJA, and MIB conceptualized and designed the study. EMC conducted all the model simulations, analysis of results, and created the figures. All authors contributed to discussing the methodology and results and to reviewing and editing the manuscript.

*Competing interests.* One of the co-authors is a member of the editorial board of Hydrology and Earth System Sciences (HESS).

*Acknowledgements.* We thank the Swiss National Science Foundation for funding this project through grant 200021_214907.



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
