# Peer review of "How well do hydrological models simulate streamflow extremes and drought-to-flood transitions?"

_EGUsphere, 2025_

## Referee Comment (RC2)

[referee-annotated manuscript omitted]

---

## Author Comment (AC1)

*Reply to reviewer's comments on the preprint egusphere-2025-781:*

**How well do hydrological models simulate streamflow extremes and drought-to-flood transitions?**

Eduardo Muñoz-Castro[1,2,3], Bailey J. Anderson[1,2,3], Paul C. Astagneau[1,2,3], Daniel L. Swain[4,5,6], Pablo A. Mendoza[7,8], Manuela I. Brunner[1,2,3]

[1]WSL Institute for Snow and Avalanche Research SLF, Davos Dorf, Switzerland
[2]Climate Change, Extremes and Natural Hazards in Alpine Regions Research Center CERC, Davos Dorf, Switzerland
[3]Institute for Atmospheric and Climate Science, ETH Zurich, Zurich, Switzerland
[4]California Institute for Water Resources, University of California Agriculture and Natural Resources, Davis, CA, USA
[5]Institute of the Environment and Sustainability, University of California, Los Angeles, Los Angeles, CA, USA
[6]Capacity Center for Climate and Weather Extremes, National Center for Atmospheric Research, Boulder, CO, USA
[7]Civil Engineering Department, Universidad de Chile, Santiago, Chile
[8]Advanced Mining Technology Centre (AMTC), Universidad de Chile, Santiago, Chile

*Correspondence to*: Eduardo Muñoz-Castro (eduardo.munoz-castro@slf.ch)

**Anonymous Referee #1**

*This manuscript presents a well-structured large-sample hydrology modeling experiment assessing the ability of four conceptual hydrological models (GR4J, GR5J, GR6J, TUW) to capture compound hydrological extremes, with a specific focus on drought-to-flood transitions. In the paper the authors examine the influence of various modeling decisions—model structure, calibration metrics, streamflow transformations, and weights—on model performance across 63 catchments in Chile and Switzerland.*

*The topic is relevant for the field of hydrology and fills some gaps in our understanding of model behavior under extremes events (drought-to-floods), which are of growing concern in the context of climate change.*

*Hence, the paper deserves to be published at HESS after some minor corrections.*

Thank you for your time and constructive feedback. Your comments have been very helpful and have contributed to improving the quality of our work. We respond to each individual point below. For clarity, *comments are given in black italics*, and our responses are given in plain blue text. Proposed additions are highlighted in red.

*General comments*

*Most of the paragraphs (e.g., L20-L38, L295-310) could benefit from some size reduction, or simply the separation of ideas. Generally speaking, one idea being introduced by paragraph would improve the readability of the text. Currently it is a bit hard to follow the paragraphs due to their size and mix of ideas together.*

Thank you for this comment. We have revised each paragraph of the manuscript to shorten its length and improve its clarity.

*Three of the four models come from the GRXJ family. This means that model structure diversity is somewhat limited. Could you please justify better this choice in the text? Also pointing out the reasoning of not including another conceptual model structure besides the GRXJs?*

Thank you for raising this point. We decided to use the GR model family to evaluate how slight differences in structure can affect the performance of models in detecting extreme events. In L173:L177 in the preprint we motivated this choice by highlighting that "GR4J, GR5J, and GR6J (with 6, 7, and 8 parameters coupled with CemaNeige, respectively) were chosen to explore how slight changes in model structure affect simulated streamflow extremes and the TUW model (with 15 parameters) was chosen to explore how more complex models, particularly with respect to the snow routine and the representation of the processes occurring in the production storage, simulate these phenomena.". To reinforce this point and discuss its limitations, we will bring this up again in Section 5.4 of the revised manuscript, where this choice is discussed in detail as follows:

To support our analysis, we tested four bucket-type hydrological models used within the hydrological modeling community (Addor and Melsen, 2019). Even though these models are at the lower end in terms of model complexity (Hrachowitz and Clark, 2017), and three of them share the same core structure, they allowed us to perform a comprehensive analysis of different model structures at a lower computational cost than when using models with more complex structures (e.g., Clark et al., 2017; Orth et al., 2015; Poncelet et al., 2017). Furthermore, previous studies have also shown that more complexity does not necessarily imply better performance (Figure 7; e.g., Li et al., 2015; Merz et al., 2022).

*The paper is dense, but could you somehow summarize better your conclusions in a maximum of three/four bullet points? I see that much can be concluded from your study, but I also think that you could benefit the readers by summarizing the main conclusions in this part rather than everything. Think about what were your hypothesis, and try to come back to them here, for example.*

We agree with the sentiment on the manuscript's density. To address this comment, we have re-structured (e.g., subsubsections in subsection 4.2 have been removed) and revised the paragraphs and sentences to limit their length. Based on this, we have revised the conclusions section to emphasize the key messages of our work, which now read as follows:

1. A satisfactory general model performance, as expressed by the KGE, does not guarantee a good performance in terms of detecting streamflow extremes and their transitions. While KGE can serve as a rough proxy for low-flow performance, it cannot for high-flows and drought-to-flood transitions. Consequently, assessments of the suitability of hydrological models for simulating extreme events and their transitions should be complemented with metrics describing extreme event detection performance such as the critical success index (CSI).

2. The most important modeling decision when it comes to simulating floods, droughts, and their transitions is the choice of a suitable model structure. Here, we demonstrate that the GR4J and TUW models have similar performance – with GR4J being slightly better at detecting floods and transitions - and adding model complexity by increasing the number of parameters does not necessarily improve the representation of extreme events.

3. The choice of the exact KGE formulation and the use of weights for their variability term to define the calibration objective function do not substantially affect the simulation of extreme events. However, a joint focus on high and low flows by equally weighting them in the objective function (referred to as HiLo in our analysis) can improve model performance without compromising its ability to capture streamflow extremes.

4. A model's performance in simulating streamflow extremes and transitions primarily depends on how well it captures streamflow timing rather than other hydrological signatures or variables such as evapotranspiration or snow water equivalent.

5. Drought-to-flood transitions are more difficult to capture in semi-arid, high-mountain, and flashy catchments than in humid low-elevation catchments.

Specific comments

*Figure 1: It is difficult to distinguish the basin boundaries in both subplots (A and B) of the figure. Maybe if you could reduce the line weight of the country boundaries in A and B, use another color for the basins and increase the figure size of subplots B, C and D.*

We agree that the catchments are not clearly distinguishable. Hence, we have changed the color of the catchment's outlet point to red and reduced the line thickness of the international borders, as well as the catchment's boundaries to improve clarity.

*L128: I think this section would benefit from this reference:*

*Clerc-Schwarzenbach, F. M., Selleri, G., Neri, M., Toth, E., van Meerveld, I., and Seibert, J.: HESS Opinions: A few camels or a whole caravan?, EGUsphere [preprint], https://doi.org/10.5194/egusphere-2024-864, 2024.*

*In their study they show that most of the time using local information (as you did) can be beneficial for model simulations. If you feel that fits, please consider inserting it.*

We appreciate the reviewer's observation, as one of our motivations for using "local" CAMELS databases was indeed based on the evidence presented in the work of Clerc-Schwarzenbach et al. (2024). We have included this reference in a discussion of forcing factors in Section 5.3, which reads as follows:

Here, we attempt to reduce this effect by (1) utilizing local meteorological products over global ones, based on the evidence that these may enhance hydrological modeling (e.g., Clerc-Schwarzenbach et al., 2024), and (2) incorporating adjustment factors to account for potential systematic biases associated with them (e.g., Hughes, 2024; Probst and Mauser, 2022). However, introducing forcing adjustment factors could compensate for some model deficiencies by modifying the inputs (e.g., Tang et al., 2023, 2025). This is somehow reflected by the high dispersion of forcing adjustment factors within each configuration (Figure S16 in the Supplementary Material). Therefore, an improvement in the spatiotemporal representation of precipitation and temperature, as well as of the potential interactions between these variables, could contribute to improved representations of compound streamflow extreme events in hydrological models.

*L329-333: I feel that this part should rather be placed in the discussion section.*

We agree that this point is interesting for discussion. However, we believe it is important to include it in the results section, as the statement directly results from that analysis. Nevertheless, following the reviewer's recommendation, this message has been reinforced in the Discussion section where it now reads:

Our comparison also highlights that the potential benefit from adjusting these choices (e.g., using other weights or other transformations) varies widely between catchments (Figure 5). This is in line with the findings of Mizukami et al. (2019), who found that the influence of weights on model performance depends on model structure and catchment characteristics. While none of the tested modifications in the objective function consistently improve the simulation of streamflow extremes across all catchments in the study domain, some of the alternative KGE formulations could improve the simulation of certain variables.

*Figure 8: The current choice of line colors and types makes it hard to distinguish among the different models. Please consider restructuring it to make it easier for readers.*

The colors have been modified to enhance the readability of the figure, and the caption for the figure has been rewritten. Thank you for this suggestion!

Now the figure (and the caption) is as below:

[Figure]

Figure 8. Example of how different hydrological fluxes and states (rows) are simulated for an observed drought-to-flood transition in the Dischma river (Switzerland) with the GR4J, GR5J, GR6J, and TUW hydrological models (colored lines) calibrated with the unweighted HiLo original KGE formulation. The shaded red and blue areas indicate periods of observed streamflow drought and flood conditions, respectively.

*Section 4.5: Start by introducing the figure, then you can make your statements. Currently it is a bit confusing the way the section is structured. Also, I see the possibility of having two paragraphs here rather than just one.*

The paragraph has been restructured and divided into two, following your recommendation. It now reads as follows:

We explore the relationship between model performance and catchment characteristics using the Spearman's rank correlation coefficient. To this end, we focus on the CSI obtained for the different types of extreme events of interest (droughts, floods, and transitions) generated with the GR4J and TUW models calibrated with the unweighted HiLo original KGE formulation (Figure 10).

Our results show that the model's capability for simulating extreme hydrological events and their transitions depends on catchment characteristics (Figure 10). Drought-to-flood transitions are more difficult to capture in semi-arid (negative correlation between aridity index and CSI), high-mountain (negative correlation between mean elevation and CSI), and flashy (negative correlation between the slope of the flow duration curve and CSI) catchments than in humid low-elevation catchments with high streamflow elasticity to precipitation. This result is generalizable to the other models and different KGE formulations tested (see Figure S14 in the Supplementary Material).

*Section 4.6: Again, please start by introducing the figure, then you can make your statements.*

To improve the clarity of the sentence and following the reviewer's recommendation, we have rewritten it as follows:

We conduct an ANOVA test to analyze the relative importance of different model parameters in detecting streamflow extremes and their transitions, whose results are presented in Figure 11 (for the extended version with rapid and seasonal transitions, see Figure S15 in the Supplementary Material).

*L472-L473: Statement repetition. This idea has already been presented.*

Thank you for bringing this to our attention. This statement has been merged with a similar one presented in L459:460 in the pre-print to avoid repetition. The new statement at the beginning of Section 5.3 reads as follows:

Our results show that model structure is the most important modeling decision for capturing extreme events and their transitions (Figure 9), which is consistent with previous studies focused on the independent analysis of extreme events (e.g., Alexander et al., 2023; Melsen and Guse, 2019; van Kempen et al., 2021).

*L501: Not Figure 10?*

In figure 10, we aim to identify, in which types of catchments the detection of extreme events works best, based on their attributes (comparison between catchment attributes and absolute CSI values). In the context of the paragraph, the idea is to highlight the role of forcing adjustment factors in explaining the variability of the CSI (i.e., the relative importance of the parameter), which may not be explicitly presented in the text. To address this, the sentence is modified as follows:

Given the relative importance shown by the forcing adjustment parameters (Figure 11), the meteorological forcings used to simulate streamflow extreme events can also have a major impact on the model's performance.

*L523-L528: This idea has already been presented in the study area. Please consider keeping it just here in the discussion.*

Thank you for this suggestion, which we will adopt. This idea will be removed from Section 3.2.1 (L177-L179 in the preprint).

**References**

Clerc-Schwarzenbach, F., Selleri, G., Neri, M., Toth, E., van Meerveld, I., and Seibert, J.: Large-sample hydrology – a few camels or a whole caravan?, Hydrology and Earth System Sciences, 28, 4219–4237, https://doi.org/10.5194/hess-28-4219-2024, publisher: Copernicus GmbH, 2024.

---

## Author Comment (AC2)

*Reply to reviewer's comments on the preprint egusphere-2025-781:*

**How well do hydrological models simulate streamflow extremes and drought-to-flood transitions?**

Eduardo Muñoz-Castro[1,2,3], Bailey J. Anderson[1,2,3], Paul C. Astagneau[1,2,3], Daniel L. Swain[4,5,6], Pablo A. Mendoza[7,8], Manuela I. Brunner[1,2,3]

[1]WSL Institute for Snow and Avalanche Research SLF, Davos Dorf, Switzerland
[2]Climate Change, Extremes and Natural Hazards in Alpine Regions Research Center CERC, Davos Dorf, Switzerland
[3]Institute for Atmospheric and Climate Science, ETH Zurich, Zurich, Switzerland
[4]California Institute for Water Resources, University of California Agriculture and Natural Resources, Davis, CA, USA
[5]Institute of the Environment and Sustainability, University of California, Los Angeles, Los Angeles, CA, USA
[6]Capacity Center for Climate and Weather Extremes, National Center for Atmospheric Research, Boulder, CO, USA
[7]Civil Engineering Department, Universidad de Chile, Santiago, Chile
[8]Advanced Mining Technology Centre (AMTC), Universidad de Chile, Santiago, Chile

*Correspondence to*: Eduardo Muñoz-Castro (eduardo.munoz-castro@slf.ch)

**Referee #2 – Dr. Wouter Knoben**

Thank you very much for your constructive feedback. Your comments have been very helpful and have contributed to improving the quality of our work. First, we address the comments the reviewer considers most important, included in the online discussion, and then the detailed comments provided directly in the PDF of our manuscript. For clarity, *comments are given in black italics*, and our responses are given in plain blue text. Proposed additions are highlighted in red.

**Comments online**

*Summary:*

*This paper presents a comprehensive analysis of the use of the Kling-Gupta Efficiency to configure conceptual models for accurate simulation of droughts, floods, and the transitions from droughts to floods. The authors use 63 basins, 4 conceptual models, 5 different KGE formulations, 3 streamflow transformations (Q, 1/Q, average of both), and 4 different weights for each KGEs' variability component. They investigate (1) the performance of the models for extreme events detection with the Critical Success Index (CSI); (2) the simulation of various hydrologic states and fluxes (streamflow, evapotranspiration, snow water equivalent, soil moisture, baseflow) with the Nash-Sutcliffe Efficiency; (3) the correlation between catchment attributes and CSI values; (4) the relative importance of different parameter values; as well as (5) the relative importance of all tested factors in terms of CSI through with an ANOVA approach. Conclusions suggest (1) that higher KGE scores for streamflow simulation do not necessarily imply good performance on other variables (such as ET or SWE) or other metrics (such as CSI and NSE); and (2) that model structure, catchment attributes and forcing data quality are key controls on our ability to accurately simulate extreme events.*

*Assessment:*

*Dear authors,*

*I have completed my review of your paper. It is comprehensive and interesting (as well as ambitious), but I believe further work is needed before this manuscript can be published. I will upload a PDF with individual comments, and attempt to summarize what I consider most important here.*

*[1] First, let me say that I recognize the sheer amount of work being presented here. However, this can be as much a weakness as a strength. Right now my feeling is that this manuscript is trying to do too much at once. The analysis is very complex, the details don't always get the attention that I think they need (I tried to highlight these instances in the PDF), and part of the methodology seems only explained in the Results sections. You are asking quite a lot of the reader throughout, and I wonder if a more streamlined and focused manuscript wouldn't present a clearer, more easily digestible message. In the end it is of course the authors' choice what to present but I wanted to point out that I found the manuscript quite difficult to follow at times due to the sheer complexity of the work being presented.*

We very much appreciate your recommendations, which we have taken into consideration and adopted to clarify the message of our manuscript. We have compiled all the observations and comments included in the revised manuscript's PDF uploaded by the reviewer and responded to each one. You can find these details below. However, the changes implemented based on the feedback received are summarized below:

1) The introduction has been modified to guide the reader to the gaps identified and how these translate into the research questions addressed in our study.

2) Figures that were difficult to understand (e.g., Figures 5 and 6) have been modified to provide a more straightforward analysis and interpretation. In addition, statistical significance test results have been included in the Supplementary Material to support the analyses presented in the manuscript.

3) The analyses/comparisons presented in the manuscript have been reduced. Now, only the different weights and KGE formulations are considered, with 'HiLo' as a fixed modeling decision. The results associated with the 'Hi' and 'Lo' cases are included in the Supplementary Material.

4) The discussion is complemented with some reflections related to forcing adjustment factors and model calibration.

*[2] Second, I think there are some technical questions/concerns that should be addressed.*

*[2a] Primarily, I think there are reasons to believe that some of the conclusions about the importance of model structure may be artifacts of improper calibration of the GR5J and GR6J models. It is my understanding that GR5J is so close to GR4J in terms of structure that it can actually become GR4J if its X5 parameter is set to 0. The GR6J case is less clear to me, but looking at the model schematics side-by-side, it should be rather close to GR4J in terms of capabilities. However, Figure 4 shows very large differences in the calibration KGE scores between GR4J on the one hand (these are higher), and GR5J and GR6J on the other (these scores are much lower). This suggests that something has not gone quite right during the calibration of GR5J and GR6J: if GR4J is strictly a subset of GR5J, than the calibration should be able to find those parameter sets that make GR5J (that currently has the much lower scores) mimic GR4J (which currently has the much higher scores). A similar argument could (should?) apply to GR6J. My first guess would be that the ranges for parameters X5 and X6 were too restrictive, but I could not find the parameter ranges that were used for calibration in the document and thus cannot confirm this. To me, this aspect of the study needs considerable attention because it affects all the analysis that comes after.*

Our concept of "satisfactory calibration" is based on the convergence of our calibration algorithm, which we acknowledge is not entirely accurate in practice due to equifinality problems. We defined a plausible range for the parameter space, large enough to avoid calibrated parameters on the edges, which has been verified. To address these points, we have included some reflection about the topic in the discussion section of the manuscript. Further details are provided later, but in short:

1) We have checked the structure and the equations of GR4J, GR5J, and GR6J and found that X5=0 does not imply GR5J being equivalent to GR4J. Similarly, X6 = 0 for GR6J does not imply equivalence to GR5J.

2) In the Supplementary Material, we included the range of parameters for the GRXJ and TUW model, as well as a brief description of each parameter and its units, to show that we explored a large enough parameter space, which means that this should not restrict the convergence of the calibration algorithm.

3) In the discussion, we include some reflections related to model calibration and the meaning of "convergence" during the optimization process.

*[2b] I'm not fully convinced of the value of adding precipitation and temperature correction factors to the calibration procedure. I think there is a real risk that this makes the calibration problem (even more) poorly constrained by increasing equifinality during parameter selection, and the benefits of adding these parameters are unclear to me. These two parameters are mainly used to support the conclusion that more accurate forcing data is helpful, but I think this is obvious. The drawback of adding these parameters (letting the calibrating procedure compensate for other shortcomings in the modelling chain by adjusting the model inputs; i.e. getting the right results for the wrong reasons) are insufficiently discussed in the manuscript.*

Thank you for your suggestions in the PDF, which guided us in improving our discussion of this methodological decision. With the incorporation of dP and dT, we aimed to avoid overcompensating for the forcings error and "let" parameters adjust the water balance – and forced their closure - through losses or gains which don't come from the precipitation input (e.g., X2 for GR4J). To summarize our answer (further details are provided later), we now discuss the implications of including these factors in the calibration and how they can compensate for some model deficiencies. Specifically, we:

1) Discuss the disagreement between the calibrated forcing adjustment factors (dP and dT) derived from sixty calibration configurations and four hydrological models for each catchment, resulting in 240 values per point, which highlights that our approach may not only correct systematic bias in forcings but could also compensate for other model deficiencies.

2) Recognize that by incorporating these adjustment factors, we may be compensating for other deficiencies.

*[2c] As far as I can tell, many (all?) of the comparisons between different setups are based on visual assessment of the figures shown in the manuscript. I believe the use of statistical tests would be more appropriate to quantify the extent to which decision X leads to different modeling outcomes than decision Y.*

We agree that some of the comparisons would benefit from statistical tests. We therefore performed some statistical tests to test the difference between the use of the unweighted original KGE formulation compared to different weights and KGE formulations. With this, we aim to reinforce the results that will be presented in the proposed updated Figure 5 (these figure is shown below as part of the specific responses to the comments made on Figure 5 of the preprint).

*[3] Finally, I think some of the statements made in the paper are too general. In particular, the first conclusion presented in Section 6 is: "Drought events are better captured by hydrological models than flood events." This is, in my opinion, too bold of a claim for the results shown in this work and needs some sort of mention about the study constraints this conclusion is valid within (number of models, types, number of basins, locations, etc.). I believe this applies to a few other statements in the paper and tried to highlight these in the PDF.*

We have reviewed our conclusions and removed some of them from the manuscript to refine them. We have also rewritten some sections of the document to acknowledge that our results are based on experiments conducted using four hydrological models. For instance, the conclusions now read as follows:

1. A satisfactory general model performance, as expressed by the KGE, does not guarantee a good performance in terms of detecting streamflow extremes and their transitions. While KGE can serve as a rough proxy for low-flow performance, it cannot for high-flows and drought-to-flood transitions. Consequently, assessments of the suitability of hydrological models for simulating extreme events and their transitions should be complemented with metrics describing extreme event detection performance such as the critical success index (CSI).

2. The most important modeling decision when it comes to simulating floods, droughts, and their transitions is the choice of a suitable model structure. Here, we demonstrate that the GR4J and TUW models have similar performance – with GR4J being slightly better at detecting floods and transitions - and adding model complexity by increasing the number of parameters does not necessarily improve the representation of extreme events.

3. The choice of the exact KGE formulation and the use of weights for their variability term to define the calibration objective function do not substantially affect the simulation of extreme events. However, a joint focus on high and low flows by equally weighting them in the objective function (referred to as HiLo in our analysis) can improve model performance without compromising its ability to capture streamflow extremes.

4. A model's performance in simulating streamflow extremes and transitions primarily depends on how well it captures streamflow timing rather than other hydrological signatures or variables such as evapotranspiration or snow water equivalent.

5. Drought-to-flood transitions are more difficult to capture in semi-arid, high-mountain, and flashy catchments than in humid low-elevation catchments.

**Comments on the PDF**

*L15-L18: I may have missed these recommendations, but I cannot recall them being so prominent in the manuscript text as this sentence suggests. Maybe they can be emphasized a bit better in the text.*

We agree that the statement may sound prominent in relation to what is presented in the manuscript (more technical analysis and not applied to risks). We have reworded the text as follows:

Ultimately, our study aims to provide insights for further model improvements targeting drought-to-flood transitions, which can support a better understanding of these compound events and identify regions prone to this hazard.

*L49-L51: Some initial thoughts based on lines 28-33 may suggest that current models don't include the necessary process granularity to model complex events and get the right results for the right reasons. These examples suggest changes to the landscape structure (e.g., landslides) or landscape properties (e.g., reduced infiltration capacity due to crust formations during prolonged drought) that are not easily captured by the current generation of models, because these models have time-invariant structures and parameters. Can we a priori expect them to do poorly?*

To the best of our knowledge, there is no demonstration that these changes can impact drought-to-flood transitions observed at the catchment outlet. Depending on the scale of interest, it might not be worthwhile to incorporate such changes explicitly into the modeling chain (maybe this will also open up an alternative research topic) because introducing time-varying parameters and structures might compensate for other uncertainties, such as the dependence of residuals on variations in the forcings. In this sense, we cannot assume a priori that model performance is affected by these potential changes without detailed information about their impact on the target variable and scale. Nevertheless, the variety of catchments in our study

could be a proxy of the impact of landscape and landscape changes on drought-to-flood transitions, which we explore in our study.

*L77: Is this true? Might be cleaner to simply state that KGE has the benefit of straightforward disaggregation of components, without diving into the question of which metric is used more often.*

With this sentence, we were referring to the popularity of KGE as a metric. We agree that the sentence could benefit from simplification. It now reads as follows:

Thanks to the possibility of disaggregating it into its three components – bias, variability, and correlation – KGE provides interpretability and diagnostic power.

*L81: This is definitely an example of a threshold, but why is it necessary to give this specific example at all? I think this invites more questions (why 0.4 and not 0.35, or 0.5, or -1?) than it clarifies.*

We agree with the reviewer that these thresholds are arbitrary. The context of this sentence points to criticism that these thresholds are usually used to justify the applicability of a model to represent, e.g., droughts and floods. For instance, the studies we cite in the previous sentence used a fixed arbitrary threshold to filter catchments with good/poor performance. To highlight this problem, we have rewritten the sentence as follow:

In these studies, despite a lack of objectivity in assessing the model's explanatory power in each catchment (i.e., benchmark; e.g., Knoben, 2024; Seibert et al., 2018), calibrations are considered successful if the KGE performance exceeds a certain threshold during both the calibration and evaluation periods (e.g., KGE > 0.4). However, there is often no explicit evaluation of how drought or flood events are represented at the event scale.

*L85-L86: I'm happy to accept that KGE has not been used extensively to evaluate consecutive extreme events, but are there other methods that have been used? If not, that would be good to indicate. If there are, some overview of what is current practice for evaluating extremes and consecutive extreme events is needed.*

In the introduction, we aimed to highlight that several studies have used hydrological models calibrated (or evaluated) with KGE to investigate droughts and floods. However, we argue that (1) there is no clear analysis of the models' ability to detect extreme events, and a "good" KGE is used as a proxy to validate the use of a hydrological model designed to study events of this type, and (2) the representation of both drought and floods, as well as their transitions, has not been jointly evaluated. To be more precise and clearer on this point, we have rewritten the summary preceding the research questions as follows:

In summary, the effectiveness of overall performance metrics - such as KGE - in evaluating the models' ability to capture streamflow extremes has not yet been thoroughly examined. Additionally, it is unclear how different modeling decisions - such as the hydrological model, objective function, and streamflow transformations - affect drought-to-flood transitions simulations. Even more, it remains to be explored which modeling choices are most suitable for capturing these compound hydrological extreme events without compromising hydrological consistency (i.e., representation of different hydrological processes or properties). To address these research gaps, we investigate the extent to which hydrological models can represent consecutive drought-to-flood transitions and the impact of model complexity and calibration choices on their representation.

*NOTE:*
*Coming back to this comment after reading the whole paper, I think this deserves more attention. The whole paper revolves around the question "can KGE (somehow) be used to configure models that accurately*

*simulate droughts, floods and the transitions between them?", and I think some more justification for asking this question is needed. Some things that I think need to be added:*

*- An overview of what is considered current best practice in evaluating floods, droughts and transitions? If there are no such practice, state that clearly. If there are, to what extent could KGE approximate these approaches (in other words, is there a reason to assume that KGE would be good for this purpose)?*

Recognizing that KGE is widely used for the calibration and evaluation of hydrological models, our manuscript aims to discuss the suitability of KGE for calibrating models to simulate streamflow extreme events, particularly drought-to-flood transitions. Then, our analysis focuses on the models' performance in capturing droughts, floods, and their transitions, challenging the assumption that a "good" KGE (or an alternative goodness-of-fit metric) is sufficient to ensure good model performance for simulating extremes. We have refined the introduction and highlighted these points to make our arguments clearer. For example, we have rewritten the summary preceding the research questions (see the answer above), and the paragraph introducing these shortcomings in the use of KGE in the context of streamflow extremes (L76-L86 in the preprint) now reads as follows:

The Kling-Gupta Efficiency (KGE), originally proposed by Gupta et al. (2009), has been one of the most popular performance metrics used in hydrology in the last decades. Thanks to the possibility of disaggregating it into its three components - bias, variability, and correlation - KGE provides interpretability and diagnostic power. It has been applied for many modeling purposes, including the analysis of streamflow extremes (e.g., Gu et al., 2023; Hirpa et al., 2018). In these studies, despite the lack of objectivity in assessing the model's explanatory power in each catchment (i.e., benchmark; e.g., Knoben, 2024; Seibert et al., 2018), calibrations are considered successful if the KGE performance exceeds a certain threshold during both the calibration and evaluation periods (e.g., KGE > 0.4). However, there is often no explicit evaluation of how drought or flood events are represented at the event scale. Thus, the suitability of KGE and alternative formulations (Gupta et al., 2009; Kling et al., 2012; Pool et al., 2018; Tang et al., 2021; Pizarro and Jorquera, 2024) or adaptations (e.g., transformations and weights; Garcia et al., 2017; Wu et al., 2025; Mizukami et al., 2019) for calibrating models aimed at studying streamflow extreme events and, in particular, consecutive extremes has not yet been sufficiently evaluated.

*- What is currently known about the four research questions listed in lines 95-100? Right now the introduction does not really go into any of that, which implies that the current state of knowledge is "we know nothing". I assume that probably isn't the case, but if it is, stating that also should be stated clearly.*

To reduce complexity in our manuscript, we have removed the second research question included in the preprint and narrowed it down to 3. Additionally, the introduction has been revised to highlight the gaps and enhance its connection to the research questions.

*L87-L88: This is a more specific set of extreme events than the introduction so far as mentioned, and it might be good to explain why the focus is here (and not eg on flood-to-drought, or flood-to-flood).*

We have focused our analysis solely on drought-to-flood transitions given their inherent asymmetry in the spatiotemporal characteristics, as well as the underlying drivers in both meteorological and hydrological perspectives, compared to flood-to-drought transitions. While the transition from drought to flood can occur within hours or days, the transitions to droughts can range from weeks to years. This asymmetry has been recently discussed by Swain et al. (2025). Consequently, drought-to-flood transitions present different challenges for, e.g., decision-makers in terms of managing the hydrological hazard that occurs on short timescales (from hours/days to weeks), which could compromise water security after the flood (e.g., poor reservoir regulation). We have included the following paragraph – updated with some publications made after the manuscript was submitted – in the manuscript to highlight this point:

As a consequence of a potential intensification of hydrological volatility in a warming climate (Swain et al., 2025), hydrological whiplash, defined as sub-seasonal transition between hydrological extremes such as droughts and floods (Hammond et al., 2025), could become more frequent and severe in the future. While the transition from drought to flood can occur within hours or days, the transition to droughts can range from weeks to years, leading to different water management challenges and reaction times for decision-makers (Hammond et al., 2025). Then, due to their inherent asymmetry in spatiotemporal characteristics and underlying drivers, as has been recently shown by Swain et al. (2025) from both meteorological and hydrological perspectives, drought-to-flood transitions can have more severe impacts than flood-to-drought transitions. Both hydrological droughts and floods are linked to meteorological conditions such as precipitation surplus/deficit or low/high evapotranspiration rates. However, Brunner et al. (2025) have shown that dry-to-wet spells are only weakly associated with drought-to-flood transitions, with a propagation rate of just 10% within a 30-day period, and that wet spells are less likely to lead to floods than dry spells are to cause droughts. Consequently, the occurrence and drivers of these compound events are not yet fully understood (e.g., Matanó et al., 2022, 2024; Brunner, 2023; Götte and Brunner, 2024; Hammond et al., 2025; Brunner et al., 2025).

*Caption Figure 1:*
- *I find these quite hard to see. Consider making them red or something, or increasing the figure size.*

    We have changed the color of the catchment's outlet point to red and reduced the line thickness of the international borders as well as the catchment's boundaries.

- *The axis label has this as P/PET.*

    The description in the figure caption has been corrected.

- *How are these calculated? I see values at around -1.2 in Fig 1d that I cannot easily interpret based on this explanation.*

    We follow the definition of Berghuijs et al. (2014), based on Woods (2009). In general terms, the index represents how in phase (or out of phase) the seasonal patterns of precipitation/streamflow and temperature are. We agree that the current description is a bit confusing, which is why we rewrote this part in the caption, which now reads as:

    For p-seasonality and q-seasonality, positive (negative) values indicate summer (winter) dominatedprecipitation or streamflow, while values close to zero indicate a uniform distribution across the year.

*L136: One key bit of information I'm missing about these catchments is how many drought-to-flood transitions this selection of (24+39=) 63 basins results in, and some statistics about these extreme events. In other words, this section needs some information about how good of a sample these basins are for the stated research questions.*

As indicated in the manuscript, our definition of thresholds for droughts and floods was based on the target of obtaining an average of roughly one event per year and catchment on average across the study domain. This results in 0.25 transition events/year (i.e., on average 1 event every 4 years). This information is included in the updated version of the manuscript in Section 3.1, which now reads:

Considering this definition and the thresholds adopted, we identified one event every four years on average for each catchment.

*Figure 3: These are quite small. Consider enlarging them.*

We enlarged the figure following your recommendation.

*L171: Is this the correct reference? I was looking for a description of GR6J in this paper but it appears to use the GR4J and MORDOR models. Should this be Pushpalatha et al., 2011 (https://doi.org/10.1016/j.jhydrol.2011.09.034)?*

Thank you for bringing this to our attention. Indeed, the reference was incorrect. In the updated version, this issue was fixed, and we now cite:

Pushpalatha, R., Perrin, C., Le Moine, N., Mathevet, T., and Andréassian, V.: A downward structural sensitivity analysis of hydrological models to improve low-flow simulation, Journal of Hydrology, 411, 66–76, https://doi.org/10.1016/j.jhydrol.2011.09.034, 2011.

*L185: What does this mean?*

With "non-conservative function", we refer to a function that enables the simulation of losses or gains of water, which doesn't come from the precipitation input, to enable water balance closure. This concept can create confusion and/or open discussion to other topics outside the scope of our study, which is why we have decided to remove the parentheses from the text.

*L199: One of the key characteristics of HBV is that it can store part of the snowmelt flux in the snowpack itself. Does TUW have this capability?*

No, the TUW model does not allow for the retention of meltwater and rainfall within the snowpack, nor does it account for the refreezing of liquid water. However, a similar structure could be retrieved by setting the water holding capacity of the snow (CWH) and refreezing coefficient (CFR) to zero. To be more explicit, the TUW description now includes the following underlined sentences:

The TUW model consists of a snow, soil, groundwater (subsurface flow), and a routing routine, similar to the HBV model (Bergström and Forsman, 1973). One of the major differences between the HBV and TUW models is found in the snow routine. The TUW model does not allow for meltwater or rainfall to be retained within the snowpack, nor does it account for the refreezing of liquid water. The snow routine partitions between liquid and solid precipitation and estimates snow accumulation and melt.

*L212-214: I have seen this approach (occasionally) in other studies with a single precipitation correction factor, but I've not seen this with two correction factors before. Given the the GR-suite of models already has the ability to deviate from the water balance through their X2 parameters, is their a risk that the calibration problem becomes (too) poorly constrained with these two correction factors added? I don't know if this will come later, but some evidence that the parameters can be effectively calibrated to a (close to) global optimum is necessary I think.*

The X2 parameter aims to adjust the water balance through a non-conservative component (i.e., artificially adding or leaking water), while the forcing adjustment factors are implemented to correct systematic biases in precipitation and temperature. The scarcity of information in high mountain regions limits the availability of observations, which could lead to these biases. For example, for CR2Met, there has been evidence of biases in precipitation (e.g., Alvarez-Garreton et al., 2018; Ayala et al., 2020) as well as for temperature (e.g., Carrasco-Escaff et al., 2023) due to, for example, the complex topography over the extratropical Andes cordillera. In short, with the incorporation of dP and dT we aimed to avoid, e.g., X2 overcompensating for the forcings error and "let it" adjust the problem of topographical vs. underground catchment. However, we recognize that incorporating additional parameters may have potential impacts on the equifinality problem,

which is somehow represented by Figure 1 (Figure S16 in the updated Supplementary Material). There, we show the calibrated forcing adjustment factors (i.e., dP and dT), combining the sixty calibration configurations and the four hydrological models for each catchment (i.e., 60 x 4 = 240 values per point), which are represented by the shapes and dispersion bars (10th and 90th percentiles). The wide variation within each catchment means that, depending on the model and the objective function configuration, the forcing adjustment factors can differ significantly. Therefore, the forcing adjustment factors do not necessarily address systematic bias (which would result in higher agreement between them), but may also be compensating for other deficiencies in the model.

To acknowledge this point, we have incorporated it into the discussion, specifically in Section 5.3, where it now reads (part of interest underlined):

Given the relative importance shown by the forcing adjustment parameters (i.e., dP and dT for all the models as well as SCF in TUW model, which seeks to correct the snow undercatch; Figure 11), the meteorological forcings can also have a major impact for detecting streamflow extremes and their transitions. Several studies have shown that errors in meteorological forcing are a key challenge in hydrological modeling (e.g., Brunner, 2023; Döll et al., 2016) due to, e.g., their large influence on the simulation of snow processes (e.g., Tang et al., 2023; Günther et al., 2019), or significant impacts on the partitioning between evaporation and runoff (e.g., Nasonova et al., 2011). Here, we attempt to reduce this effect by (1) utilizing local meteorological products over global ones, based on the evidence that these may enhance hydrological modeling (e.g., Clerc-Schwarzenbach et al., 2024), and (2) incorporating adjustment factors to account for potential systematic biases associated with them (e.g., Hughes, 2024; Probst and Mauser, 2022). However, introducing forcing adjustment factors could compensate for some model deficiencies by modifying the inputs (e.g., Tang et al., 2023, 2025). This is somehow reflected by the high dispersion of forcing adjustment factors within each configuration (Figure S16 in the Supplementary Material). Therefore, an improvement in the spatiotemporal representation of precipitation and temperature, as well as of the potential interactions between these variables, could contribute to improved representations of compound streamflow extreme events in hydrological models.

[Figure]

Figure 1: Forcing adjustment factors calibrated for each catchment for (a) Chile, and (b) Switzerland. The error bars are associated with the 10th and 90th percentiles across configurations (i.e., 60 x 4 models) per catchment, while the central shape is the 50th percentile for each catchment (i.e., 63 = 24 Chile + 39 Switzerland).

*L214: How does this interact with the PET estimation procedure? Presumably PET was derived before model calibration, but I'm not entirely sure if it makes theoretical sense to first derive a key model input from temperature, and then add a temperature-correction parameter because we think the T-series may be flawed. Was this considered during the experimental design?*

Considering that the temperature series is modified in each iteration during calibration, PET is estimated using the new temperature time series based on Oudin's formula during each iteration. This was not mentioned in the manuscript, but is now included in the methodological description as follows:

The parameters of each of the four model structures, as well as the forcing adjustment parameters introduced, were calibrated using daily streamflow records and the Shuffled Complex Evolution global optimization algorithm (SCE-UA; Duan et al., 1992) on the period between 2000-2020. This calibration period was defined to capture the current hydroclimatic conditions in the modeling setup. Considering the temperature adjustment parameter, potential evapotranspiration was recalculated in each iteration during calibration to ensure consistency between those variables.

*L216: I'm missing some info about the parameter ranges used for calibration.*

*NOTE:*
*Coming back to this, this is of particular interest for the X5 parameter (see later comments). This information must be added.*

We now include the parameter ranges used for calibration in Tables S1 and S2 in the supplementary material for the GRXJ and TUW models, respectively, along with a brief description of each.

*L222: "infinity"*

Thank you for bringing this to our attention. The word has been corrected in the updated version of the manuscript.

*L234: It would be good to connect/contrast this to the approach of Mizukami et al. 2019 (https://hess.copernicus.org/articles/23/2601/2019/).*

We agree that this study is relevant to our discussion. We have rewritten some ideas included in the manuscript (e.g., L491:L494 in the preprint) :

Our comparison also highlights that the potential benefit from adjusting these choices (e.g., using other weights or other transformations) varies widely between catchments (Figure 5). This is in line with the findings of Mizukami et al. (2019), who found that the influence of weights on model performance depends on model structure and catchment characteristics. While none of the tested modifications in the objective function consistently improve the simulation of streamflow extremes across all catchments in the study domain, some of the alternative KGE formulations could improve the simulation of certain variables.

*L236: It should be noted that this paper investigates the behaviour of 1/Q with NSE and makes recommendations for that, and that there is therefore an assumption in the current paper that their suggested approach holds for KGE.*

Yes, we assume Pushalatha et al.' (2012) recommendation to be valid to avoid issues in the computation of the KGE transformed flow values when simulated or observed flows equal zero. This recommendation has also been adopted in other studies using KGE (e.g., Garcia et al., 2017; Knoben et al., 2020; Pizarro and Jorquera, 2024). However, we understand that the current formulation of the sentence could be misleading, which is why we have rewritten it as follows:

Following the recommendations from previous studies, we add a constant equal to 1% of the mean streamflow for the low-flow transformation (Low; i.e., using 1/Q), to avoid zero-flow problems (e.g., Pushpalatha et al., 2012; Garcia et al., 2017; Knoben et al., 2020).

*L249-L250: It's not entirely clear to me what NSE offers here that KGE doesn't in terms of recognition and ease of interpretation. Can this be clarified?*

*I also don't quite understand what "compare results independently" refers to. Independent from what?*

*Finally, we know (1) that models calibrated for one metric (in this case KGE) don't necessarily perform well on another metric (in this case NSE), and (2) that NSE and KGE have no clear 1-to-1 mapping, and instead relate to one another through a function that includes the coefficient of variation of the observations as a term. In other words, a KGE in a basin with CVobs=1 will give a different NSE than that same KGE in a basin with CVobs=0.5 or CVobs=2.*

*I think all three questions deserve some attention individually, but my main question is probably why this switch in metrics is necessary. This paper already does so many things that the reader needs to work hard to keep up, and I don't really see what switching to NSE offers that compensates for introducing even more complexity in the methodology.*

The use of NSE provides us with a KGE-independent and alternative evaluation through a well-known and easily interpretable metric. By independent, we mean a metric other than KGE and, likewise, the definition we adopt for it. Consequently, if we decide to use KGE as a comparison metric, we will have at least 5 different alternatives (i.e., each KGE formulation). Then, the analysis should be in terms of cross-comparison of results (e.g., simulations obtained with the original KGE assessed in terms of KGE_np). We evaluated this, but it made the results even more complex to interpret.

In Figure 2, you can see this cross-comparison, which shows the change in the performance of a calibrated model with a specific formulation of KGE as an objective function (reference) when evaluated against another KGE formulation (alternative). The figure shows that, in general, for the GR4J and TUW models, the variations are close to zero and with low dispersion between basins, except in KGE_np where the reference stands out. However, when analyzing the other GR models, greater dispersion among catchments in the results is observed. We will include this figure in the supplementary material and provide some additional explanations in the discussion.

Therefore, despite the limitations highlighted by the reviewer, we decided to pursue NSE as an alternative metric to achieve an independent evaluation. However, the same analysis could have been performed with other metrics such as RMSE or PBias.

[Figure]

Figure 2: Cross-comparison between model configurations. Change in performance during the validation period of a calibrated model with a specific formulation of KGE as an objective function (reference) when evaluated against another KGE formulation (alternative).

To avoid confusion and improve clarity: (1) we have removed the analyses presented in the manuscript associated with NSE (now only presented in Supplementary Material), and (2) we have rewritten the first sentences in Section "3.3.1 Overall performance and hydrological consistency" as follows:

We compute the Nash-Sutcliffe Efficiency for different variables, including high- and low-flows (i.e., Q and 1/Q), snow water equivalent (SWE), soil moisture (SM), and actual evapotranspiration (ET). We use the NSE because it enables us to compare results independently of the KGE configuration across the different calibration experiments tested here, as well as for its interpretability and popularity within the hydrological community (Melsen et al., 2025).

*L260-L261: I assume this this meant in a temporal sense (e.g. the simulated "flood window" needs to overlap at least 50% with the observed one). Is there also a magnitude component (e.g. that a simulated peak is at least within x% of the observed one, or that a simulated drought is no more than Y% different from the observations)?*

In the manuscript, we refer only to the temporal overlap window, but we do not discriminate the performance of the model representing the specific characteristics of each event (e.g., cumulative deficit during the drought period, flow peak, etc.). Considering that each streamflow series (i.e., observed and simulated) has its own unique drought and flood thresholds, our objective is to evaluate the models' ability to capture these events rather than the characteristics (which may be even more restrictive). For instance, in Figure S18 in the supplementary material (note that the numbering will change in the updated version), we assessed the model's performance in terms of the flow peak bias, showing results on average close to zero for the hits. To mention this explicitly, we propose incorporating the following text at the end of Section "3.3.2 Detection of streamflow extreme events":

In short, we aim to evaluate the models' ability to capture streamflow extremes and their transitions rather than their characteristics (which may be even more restrictive). Therefore, we do not discriminate the performance of the model representing the specific characteristics of each event (e.g., cumulative deficit during the drought period, flow peak, etc.).

*Also, is there some justification or sensitivity analysis of this 50% choice? it seems fairly arbitrary to me and it would be good to add some support (or evidence that the conclusions are insensitive to it).*

As you pointed out, the overlap tolerance threshold is defined arbitrarily. To address this concern, a sensitivity analysis was performed, yielding the results presented in Figure 3 in this response to the reviewers, which shows that the more restrictive the overlap, the worse the performance of the models in detecting extreme events. This result is more pronounced for droughts compared to floods and transitions because droughts, by definition, have a longer duration and, therefore, may be more sensitive to the defined overlap (i.e., there is a wider time window, of weeks or months, where a lag may exist, unlike floods, which only happen in days). Considering that the detection of transitions is generally influenced by the performance of flood detection models (lower CSI associated with this type of event compared to droughts), the sensitivity of models detecting transitions depends more on floods, which is shown in Figure 3c consistently for all models (i.e., results for transitions are similar to those for floods). Figure 3 will be included in the Supplementary Material and referenced in the manuscript to complement our statement related to the sensitivity in the drought and flood thresholds as follows (underlined text):

The improvement in drought detection when the threshold is relaxed can be explained by the fact that models generally struggle during more extreme hydrological drought periods (e.g., Bruno et al., 2024), which are relatively less frequent if the threshold is lowered. Similar results are obtained when the overlap window used to identify the hits is modified (Figure S3).

[Figure]

Figure 3: Performance of the GR4J, GR5J, GR6J and TUW models (in the rows) in detecting a) droughts, b) floods, and c) transitions, according to different overlap thresholds (different colors) used for the identification of streamflow extreme events. For each type of extreme event and hydrological model, the results are compared according to different formulations of KGE (unweighted and HiLo) used as objective functions (x-axis).

*L261-L262: Another figure (or addition to the first) to highlight for both droughts and floods how far away a simulated event can be to still be considered a "hit" could be helpful to some readers.*

As we are trying to highlight the need to evaluate event detection as an alternative to traditional analyses, we believe that the figure above answers the reviewer's questions. However, for the adopted threshold configuration, the supplementary material presents the performance of the models with respect to bias.

*L295-L296: Adding some citations to support that this is indeed often done would be good.*

We now provide one example of droughts (Lema et al., 2025), floods (Cinkus et al., 2023), and both (Zhao et al., 2025), where authors have used KGE to assess the accuracy and reliability of the models' simulations aimed at representing streamflow extremes. This sentence now reads as:

Model performance described by general metrics such as KGE during calibration and evaluation is often used as a proxy for how well a model represents streamflow properties such as extreme events (e.g., Lema et al., 2025; Cinkus et al., 2023; Zhao et al., 2025).

*L301: Given the example between the brackets (CSI [0.23,0.79] for GR4J, [0.21,0.74] for TUW), is this word meant to indicate that there are also many cases where the KGE seems a poor indicator of a model's ability to simulate droughts? If so, I would suggest to make that explicit. If not, than the follow-up question is why CSI values that feel low to me (capturing only a quarter of the events doesn't seem particularly outstanding) can support this statement that the KGE can be a useful proxy for drought simulation capability.*

We agree that the statement may be confusing, which is why we have rewritten it as follows:

While the overall performance described by the KGE can potentially be a useful proxy for a model's performance in capturing droughts for some catchments (e.g., CSI ranges from 0.23 to 0.79 for GR4J and from 0.21 to 0.74 for TUW), it is not generalizable to floods and transitions or to all the models tested here.

*L302: I notice that the pattern for GR5J and GR6J is very different than that of GR4J and TUW. How do those results affect the "KGE can potentially be a useful proxy for a model's performance in capturing droughts" statement? Does it need to be updated with a "... for specific models" clause?*

We have rewritten some ideas in the paragraph to emphasize that the results are model-dependent. We proposed the following modification:

Our comparison clearly shows that model performance varies across catchments and model structures for both the KGE and CSI (Figure 3). While the overall performance described by the KGE can potentially be used as a proxy for a model's performance in capturing droughts for some catchments (e.g., CSI ranges from 0.23 to 0.79 for GR4J and from 0.21 to 0.74 for TUW), it is not generalizable to floods and transitions or to all the models tested here. Therefore, a high KGE does not necessarily imply a high CSI for these two types of events. For droughts and transitions (top and bottom panels, respectively), GR4J and TUW show comparable results both in terms of the median KGE (e.g., 0.85 and 0.80 in calibration) and CSI (e.g., 0.59 and 0.51 for droughts) as well as their variability (dispersion bars), but the model performance decreases when GR5J and GR6J are analyzed. For floods (middle panel), the GR4J outperforms the other models both in terms of general performance (median KGE is 0.06-0.11 and 0.04-0.18 larger than in other models for calibration) and the detection of extreme events (median CSI is 0.03-0.13 larger than in other models depending on the type of event).

*L303: It also doesn't seem to imply this for the GR4J and TUW cases mentioned on line 300-303. The KGE scores for droughts are relatively constrained and higher for these models than the other two GR models, but the CSI covers a very wide range: i.e., high KGE does not guarantee high CSI - the only difference seems to be that for droughts _in some cases_ higher KGE values coincide with higher CSI values, but it's by no means a given.*

Thank you for pointing out this opportunity for improvement in the presentation of our results. Considering that this recommendation is aligned with the following one, we will fully respond there.

*L304: I'm not sure if this figure is the clearest way to convey these results. It's clean, but it also simplifies all the patterns into two statistical measures that don't necessarily represent the underlying relationship (KGE vs CSI) that the crosses imply. In my opinion this cross of 10-90th percentile values is most appropriate to represent a point cloud that uniformly covers the area delineated by the cross. I expect that this is not the case, and that scatter plots of all KGE vs CSI cases will show more pattern than just a blob.*

*Given that there is a lot of unused whitespace around this figure, I would strongly recommend to (1) enlarge or separate the subplots into larger/more plots to make space for one of the following changes: (2) replace these crosses with the actual data as scatter plots, or plot the actual data with high transparency underneath the crosses (or do something else along these lines, of course).*

*I think this will strengthen the arguments made in this paper compared to what's currently being shown. I do not think these summary plots are the best way to support the text and conclusions in Section 4.1.*

We appreciate your recommendations. We have modified the figure and separated the information into 3x4 panels (rows: droughts, floods, transitions, columns: models), including the points associated with the KGE-CSI pairs. With this new configuration, our key messages – i.e., (i) a good KGE does not guarantee a good CSI, (ii) overall performance capturing droughts is better than for floods and transitions, and (iii) comparable results between GR4J and TUW, while there is a drop in performance of GR5J and GR6J compared to GR4J - are conveyed more clearly and effectively.

Now the figure (and the caption) looks as shown below:

[Figure]

Figure 4. Comparison between the Kling-Gupta Efficiency (KGE) for the calibration period and the Critical Success Index (CSI) for droughts, floods, and transitions, based on the simulations with the models (a) GR4J, (b) GR5J, (c) GR6J, and (d) TUW calibrated with the unweighted original KGE formulation as the objective function. The dispersion bars are associated with the 10th and 90th percentiles across catchments, while the central shape is associated with the 50th percentile. Circles with transparency show results for each catchment.

*L311-L314: I'm missing a section that comes before the KGE-vs-CSI relationship, namely one that assess how well the different models are calibrated at all. Looking at figure 4 it seems to me that the KGE scores for GR4J and TUW are relatively narrow distributions, whereas those for GR5J and GR6J are much wider.*

*Skimming the documentation of the GR-suite (https://cran.r-project.org/web/packages/airGR/airGR.pdf) it seems to me that GR5J should be rather close in capabilities to GR4J, to the extent that with the right parameter value for X5 (i.e., X5 = 0), GR5J effectively becomes GR4J. I'm finding it more difficult to find if GR6J can be simplified into GR4J with carefully chosen parameters.*

*However, the fact that GR4J is calibrated to a relatively narrow range of higher KGE scores, whereas GR5J obtains a much wider and overall lower range of KGE scores, implies that something during calibration has not gone quite right. These higher KGE scores can clearly be obtained by a model structure with X5=0, and that the optimisation routine didn't find this suggests that either that parameter range used for X5 was overly restrictive, or that the calibration didn't properly converge. A similar argument may apply to the GR6J case. I believe this needs to be investigated and explained. Otherwise the GR5J and GR6J results may need to be removed from the paper.*

We appreciate your question, which has led us to reflect on the differences and similarities between the GRXJ models from a perspective more closely linked to their mathematical implementation. In summary, we have checked the structure and the equations of GR4J, GR5J, and GR6J and found that X5=0 does not imply GR5J being equivalent to GR4J (see detailed explanations below). Furthermore, we explored a large range of values for the X5 and X6 parameters, as well as for the other parameters in the GRXJ models (see Table S1) and for the TUW model (see Table S2), which means that this should not restrict the convergence of the calibration algorithm.

Below, we present our reflections on this matter and our responses to the reviewer.

1) Regarding model structures:

The parameters X2 and X5 are related to the water exchange function that calculates a semi-potential exchange flux F applied to the 2 (or 3 for GR6J) routing branches. This function aims to represent the exchange between topographical catchments (Pushpalatha et al., 2011). In GR4J, F is defined as shown in Eq. (1), while in GR5J, its formulation is modified to that presented in Eq. (2), where R is the routing store level.

$$F_{GR4J}(t) = X2 \cdot \left(\frac{R(t-1)}{X3}\right)^{3.5} \tag{1}$$

$$F_{GR5J}(t) = X2 \cdot \left(\frac{R(t-1)}{X3} - X5\right) \tag{2}$$

This change in the formulation of F in GR5J (i.e., linear relationship instead of a power-law) makes it impossible to recover the structure of GR4J when X5 is set to 0. F coincides only when the ratio between R and X3 is equal to 1, which, although it could potentially occur at some time step, is unlikely to remain constant over time due to its construction, or when X2 is set to 0 (i.e., no exchange function). This modifies the direct branch output and the output of the routing store branch (Qd and QR in airGR notation and for GR4J and GR5J).

In GR6J, an exponential store, which is a bottomless reservoir whose water content "Exp" can be negative (i.e., water can enter or leak out of the system), is added to the GR5J structure. The output from this reservoir

"$Q_{Rexp}$" is computed with Eq. (3). Considering that X6 cannot be equal to 0 because Eq. (3) becomes undefined, there will always be a positive/negative flux from this storage, which makes it impossible to recover the same structure as GR5J by trying to define X6 as very close to 0 (which can cause the completely opposite effect).

$$Q_{Rexp}(t) = X6 \cdot \log\left(1 + \exp\left(\frac{Exp}{X6}\right)\right) \tag{3}$$

2) About the optimization process:

The models were calibrated using the SCE-UA algorithm configured with 10 complexes, a tolerance of 10e-3, and a maximum number of 5,000 iterations. We verified the convergence of the optimization algorithm. Consequently, in terms of optimization, we can say that the calibration was successful. However, we understand that this also involves uncertainty, and discussing a single and unique global optimum opens up the discussion to other topics, such as equifinality, which is beyond the scope of this work. We consider the calibration successful when the algorithm converges to an optimum. We also extracted all simulations around 0.01 of the optimum OF value (i.e., OF – 0.01) and compared the level of agreement of the set of parameters obtained, which is presented in Figure S11 in the supplementary material.

We define our parameter search range in such a way as to avoid restricting it or having values very close to the limits of the space. We defined our range of parameters based on our previous experience using these models (e.g., Araya et al., 2023; Muñoz-Castro et al., 2023). Figure 4 shows all the parameters obtained during calibration in this study in their normalized space (i.e., for parameter θ, θnorm = (θ - θmin)/(θmax - θmin)). There, it can be observed that, despite covering the entire defined parameter space, in general the calibrated parameters are far from their boundaries.

To acknowledge this, we have included the following discussion in the Section "5.4 Limitations and recommendations for future work":

To support our analysis, we tested four bucket-type hydrological models used within the hydrological modeling community (Addor and Melsen, 2019). Even though these models are at the lower end in terms of model complexity (Hrachowitz and Clark, 2017), and three of them share the same core structure, they allowed us to perform a comprehensive analysis of different model structures at a lower computational cost than when using models with more complex structures (e.g., Clark et al., 2017; Orth et al., 2015; Poncelet et al., 2017). Furthermore, previous studies have also shown that more complexity does not necessarily imply better performance (Figure 7; e.g., Li et al., 2015; Merz et al., 2022). These models have been calibrated based on daily streamflow records, assuming that the numerical convergence of the optimization algorithm ensures (to some extent) a successful calibration process. However, it is important to acknowledge that potential compensations for biases in meteorological forcings or model deficiencies can make the "optimal" parameter sets less identifiable (e.g., Clark and Vrugt, 2006; Vrugt et al., 2005; Beven, 2025). Here, we explore the (dis)agreement between the optimal parameters for each configuration (Figure S17 in the Supplementary Material), showing overall agreement indices of around 0.5 (i.e., the parameters have a range of variation of approximately 50% of the parameter space). This highlights the need to incorporate, for example, hydrological variables such as SWE or ET, to (i) complement model assessment, (ii) better define the parameter exploration range (Figure S15 in the Supplementary Material), and (iii) lead to parameter sets that ensure reliability and fidelity in representing hydrological processes.

[Figure]

Figure 4: Calibrated parameter values normalized according to their range of variation. Each boxplot contains 60 sets of parameters for each of the 63 basins (i.e., 60 x 63 = 3780 values).

*L316: I find this section a bit difficult to follow. I tried to outline where I struggled in extra comments below, but it might be worthwhile to have some back and forth with colleagues who are not on the author team to ensure the points the authors want to make come across as well as they can. I think it mainly originates in me struggling a bit to wrap my head around the relative comparisons shown in Fig 5 and how those results should be interpreted.*

*Also, is this section purely based on visual assessment of the boxplots shown in Figures 5 and S3, S4? Given the visual similarities I think the use of formal statistical tests would be more appropriate.*

We have restructured the section to make it easier to read and to highlight the most important points. In summary, we (1) reduced the number of subsections, (2) focused our analysis mainly on KGE formulations and the use of weights, while the HiLo transformation is used to perform some of the comparisons shown in the manuscript, (3) simplified and updated Figures 5 and 6, and (4) included the statistical test applied to

the results in the Supplementary Material. Further details are provided in the answers to the specific questions regarding the interpretation of Figure 5.

*L320-L324: This might be good to compare to the findings from Mizukami et al., 2019 (https://hess.copernicus.org/articles/23/2601/2019/). Maybe this comes later(?).*

These results are compared with the work of Mizukami et al. (2019) in the discussion section of the manuscript, where the following can be read:

Our comparison also highlights that the potential benefit from adjusting these choices (e.g., using other weights or other transformations) varies widely between catchments (Figure 5). This is in line with the findings of Mizukami et al. (2019), who found that the influence of weights on model performance depends on model structure and catchment characteristics.

*L330: Perhaps slightly more accurate to phrase this as "is not consistently beneficial and may even be detrimental, as ..."*

Following the reviewer's recommendation, we have rephrased the sentence as follows:

These results highlight that, in the context of a large-sample study, weighting the variability term of the KGE does not consistently enhance model performance in detecting streamflow extremes and their transitions (median difference is centered around 0) and may even be detrimental.

*L334-L335: I'm not sure if I fully follow this argument. As I understand things shown in Figure 5b:*
*- Fig 5b1: for droughts, there are no real detectable differences between Hi, Lo, and HiLo*
*- Fig 5b2: HiLo leads to better flood simulation than Lo*
*- Fig 5b2: HiLo leads to similar flood simulation as Hi*
*- Fig 5b3: no real detactable differences between Hi, Lo and HiLo*

*Is this really a good ground to recommend HiLo for modelling drought to flood transitions as Line 332 does? Wouldn't using just Hi give the same results?*

We have modified the figures (and the corresponding section) to improve clarity and readability. In summary, we did the following modifications:

1) Now, instead of having three subsections, we have condensed everything into one, limiting the presentation of results to the analysis of the added value of using different KGE formulations and weights. This is because there is already literature demonstrating that the use of a HiLo-type combination allows for a good compromise in the representation of high and low flows, but also, given our results, does not compromise the ability to capture extreme events. Consequently, we have included a paragraph at the beginning of the results section that reads as follows:

   The results presented here are based on the simulations with the HiLo (i.e., 0.5*KGE(Q) + 0.5*KGE(1/Q)) configuration, unless specific cases where all 60 configurations per catchment were used (e.g., ANOVA tests). This is considering that our results are consistent with other studies, which have shown that the use of such approaches enables a good compromise in simulating both low and high flows (e.g., Garcia et al., 2017; Thirel et al., 2024; Lema et al., 2025). The results for the alternative streamflow transformations are presented in the Supplementary Material.

2) We propose updating Figures 5 (as shown below) and removing Figure 6 (results now only included in the Supplementary Martial) from the manuscript. Now, we compared results only with the unweighted original (HiLo) KGE formulation (reference). Note that this reference is the same case

shown in Figure 4. Then, as one of our research questions relates to the suitability of KGE, we now present the different formulations compared with our reference. There, it can be seen that the use of alternative formulations does not report consistent changes in the detection of extreme events (medians centered on zero; see new Figure 5).

3) Statistical tests have been applied to assess the statistical significance of the differences between the results associated with different KGE formulations. Results are included in the Supplementary Material.

[Figure]

Figure 5. Difference in the CSI for GR4J simulations using model calibrations with unweighted original KGE (reference) versus different weights and KGE formulations (alternative) for (a) droughts, (b) floods, and (c) transitions. Values above (below) 0 indicate better (worse) performance of the reference compared to the alternative.

In response to the reviewer's question associated with Figure 5 presented in the preprint, the recommendation of HiLo is supported by Figure 6 (preprint). Here we present an adapted version of Figure 6 to focus our discussion on the results associated with NSE(Q) and NSE(1/Q). From it, we can draw the following conclusions:

1) By using HiLo to calibrate the hydrological model (reference), model performance simulating high-flows (NSE(Q)) has comparable results to the calibration without any streamflow transformation. However, model performance simulating low-flows (NSE(1/Q)) is clearly superior when calibrating models with the HiLo configuration compared to Hi (Figure 5a, upper row).

2) Using HiLo to calibrate the hydrological model (reference), improves model performance for high-flow simulations compared to using only low transformations (Lo), while still showing similar results for low-flow simulations (Figure 5a, lower row).

3) When comparing the unweighted HiLo version of each KGE formulation to the original, there are no pronounced differences regarding model performance in simulating high and low-flows (NSE(Q) and NSE(1/Q) in Figure 5b respectively).

[Figure]

[Figure]

Figure 5: Adapted version of Figure 6 included in the preprint.

Then, combining these results with those presented in Figure 5 (pre-print), where we showed that, in the context of large-sample studies, the use of weights applied to the variability term of the KGE is not meaningful for the model performance detecting droughts, floods, and their transitions, we can conclude that HiLo can provide us with more hydrological consistency. Wehope that the updated figures and text will make it easier to interpret our results.

*L336: I think we know that models calibrated for streamflow do not necessarily perform well on other variables. Is there some a priori reason to assume that using different streamflow transformations or KGE formulations (next section) would somehow change this?*

*It's not in any way bad to investigate this, but given the complexity of this paper I wonder if this investigation adds enough to warrant including it. If there are no immediate reasons to assume these transformations would somehow improve the simulation of non-streamflow variables then perhaps the paper might be streamlined by removing those extra variables.*

The incorporation of an alternative variable to Q aims to identify any configuration that stands out in terms of process representations. However, we agree with the reviewer that the manuscript needs to be simplified to improve and clarify the main messages. We have modified the figure and now only present NSE(Q), NSE(1/Q), and NSE(SWE).

*L337: Just noting here that the title of Figure S6 mentions KGE but the axis label and caption below if use NSE.*

*NOTE:*
*Returning to this after reading the whole paper, I think the part that was unclear to me was the switch to the NSE metric, and, w.r.t. the next comment, some text in the SI may be necessary to help the reader understand what is happening.*

We have removed the analyses presented in the manuscript associated with NSE (now only presented in the Supplementary Material). Additionally, we have supplemented the descriptions of some figures and tables presented in the supplementary material to improve their readability (see the following response).

*L338: I find the SI quite hard to interpret without any guidance. It might be worthwhile to add some text there to support this argument for the interested reader.*

To improve the readability of the supplementary material, we have included a brief introductory paragraph for some figures/tables to explain what is shown there when they are not a pure generalization of the results presented in the main manuscript. For example, in Figure S6 in the first submission, we have included the following description (similar for other figures as S8 and S9):

Figure S7 shows the performance of the hydrological models (x-axis) calibrated with different KGE formulations (colors). Here, we only considered the unweighted HiLo configuration (i.e., each boxplot contains 63 values, one per catchment). Model performance was evaluated using the NSE calculated for the daily series of streamflow, targeted at high (Q) and low flows (1/Q), snow water equivalent (SWE), actual evapotranspiration (ET), and soil moisture (SM).

[Figure]

Figure S6: NSE calculated for the (a) calibration and (b) evaluation periods for different hydrological variables shown in the rows. From top to bottom: daily high and low flows (i.e., Q and 1/Q), snow water equivalent (SWE), actual evapotranspiration (ET), and surface soil moisture (SM). The modeling results are associated with the GR4J, GR5J, GR6J, and TUW hydrological models (x-axis) calibrated using the unweighted HiLo configuration and different KGE formulations (colors). The dashed black lines indicate the optimal value for the assessed metric.

*L350: I find this section difficult to follow. The complications of assessing a KGE-calibrated model for different variables with a different metric is hard to get my head around (see earlier comment about the potential complications with introducing NSE as part of the evaluation methodology). Is bringing NSE into this really necessary or can this section simply use KGE as well in an attempt to keep things a bit manageable for the reader?*

The use of NSE offers a KGE-independent and alternative evaluation using a well-known and easily interpretable metric. However, to enhance readability and clarity, we have removed this analysis from the

main text and now include it in the Supplementary Material, where it serves to show the performance of our simulations.

*L369-L371: In an earlier comment I noted that there may be reasons to believe that the calibration of GR5J and GR6J has not been so successful. Given that, I do not think that this statement is currently supported by the experiment as is.*

For each calibration, we verify the convergence of the optimization algorithm. However, we understand that this does not mean obtaining an optimum that ensures the uniqueness of the solution. We have included some thoughts on this in the discussion. Then, in the Section "5.4 Limitations and recommendations for future work" it now reads:

To support our analysis, we tested four bucket-type hydrological models used within the hydrological modeling community (Addor and Melsen, 2019). Even though these models are at the lower end in terms of model complexity (Hrachowitz and Clark, 2017), and three of them share the same core structure, they allowed us to perform a comprehensive analysis of different model structures at a lower computational cost than when using models with more complex structures (e.g., Clark et al., 2017; Orth et al., 2015; Poncelet et al., 2017). Furthermore, previous studies have also shown that more complexity does not necessarily imply better performance (Figure 7; e.g., Li et al., 2015; Merz et al., 2022). These models have been calibrated based on daily streamflow records, assuming that the numerical convergence of the optimization algorithm ensures (to some extent) a successful calibration process. However, it is important to acknowledge that potential compensations for biases in meteorological forcings or model deficiencies can make the "optimal" parameter sets less identifiable (e.g., Clark and Vrugt, 2006; Vrugt et al., 2005; Beven, 2025). Here, we explore the (dis)agreement between the optimal parameters for each configuration (Figure S17 in the Supplementary Material), showing overall agreement indices of around 0.5 (i.e., the parameters have a range of variation of approximately 50% of the parameter space). This highlights the need to incorporate, for example, hydrological variables such as SWE or ET, to (i) complement model assessment, (ii) better define the parameter exploration range (Figure S15 in the Supplementary Material), and (iii) lead to parameter sets that ensure reliability and fidelity in representing hydrological processes.

*L372: This is a distinctly subjective opinion. TUW may have more calibrated parameters, but in terms of non-linearity I wouldn't be surprised of the GR-suite of models can create a much wider variety of outputs than TUW can. Seeing how there is no objective definition of "model complexity", I would recommend to stay away from statements about relative complexity if these cannot be easily supported.*

We agree that it is better to be precise in our terminology in this sentence, which is why instead of referring to "model complexity" in that sentence, we now refer to the differences in the number of parameters.

*L374: Where can the reader find this information? I don't think this is visible in Fig. 7.*

This information can be found in Figure S11 in the supplementary material. We mentioned this initially in L378 in the pre-print, but after this comment, we think the reference to Figure S11 should be made where the reviewer points out the missing information to enhance readability. To clarify, now it reads as (numeration has been updated):

If model structures are compared for Switzerland and Chile (see Figure S12 in Supplementary Material), the same conclusions can be drawn in terms of model performance, with comparable results between GR4J and TUW and lower performance of the GR5J and GR6J. However, the detection of extreme events is more challenging in catchments located in Chile compared to those located in Switzerland, with differences in the median CSI between countries being around 0.23, 0.02, and 0.13 for droughts, floods, and drought-to-flood transitions, respectively. In summary, the GR4J and TUW models seem to be the models best suited for simulating droughts and floods among the models considered in this study.

*L389: Where can these be found?*

The results of the ANOVA are presented in Figure 7. Now we include the reference to the figure in the sentence to enhance readability.

*L390: Just noting again that if GR5J and GR6J are not successfully calibrated, this conclusion may not be valid.*

Our calibration has converged on the number of iterations defined in the optimization process. Then, considering convergence as evidence of satisfactory calibration, our conclusion is still valid. However, in the limitations section of the discussion, we have included some reflections on this topic, which align with another comment made by the reviewer regarding the importance of the calibration algorithm and its impact on the results obtained. Thus, we have included the following text in the Section "5.4 Limitations and recommendations for future work":

To support our analysis, we tested four bucket-type hydrological models (Addor and Melsen, 2019). These models are at the lower end in terms of model complexity (Hrachowitz and Clark, 2017), and three of them share the same core structure, however, they allowed us to perform a comprehensive analysis of different model structures at a lower computational cost than when using models with more complex structures (e.g., Clark et al., 2017; Orth et al., 2015; Poncelet et al., 2017). Furthermore, previous studies have also shown that more complexity does not necessarily imply better performance (Figure 7; e.g., Li et al., 2015; Merz et al., 2022). These models have been calibrated based on daily streamflow records, assuming that the numerical convergence of the optimization algorithm ensures, a successful calibration process to some extent. It is important to acknowledge that potential compensations for biases in meteorological forcings or model deficiencies can make the "optimal" parameter sets less identifiable (e.g., Clark and Vrugt, 2006; Vrugt et al., 2005; Beven, 2025). Here, we explore the (dis)agreement between the optimal parameters for each configuration (Figure S17 in the Supplementary Material), showing overall agreement indices of around 0.5 (i.e., the parameters have a range of variation of approximately 50% of the parameter space). This highlights the need to incorporate, for example, hydrological variables such as SWE or ET, to (i) complement model assessment, (ii) better define the parameter exploration range (Figure S15 in the Supplementary Material), and (iii) lead to parameter sets that ensure reliability and fidelity in representing hydrological processes.

*Figure 8: If this model produces no baseflow, where does its flow during the dry period in the top plot come from?*

The figure has been updated. When generating it, we had forgotten to add the output from the exponential storage (QRExp) to the output from the routing storage (QR) when estimating the base flow in the GR6J model. This calculation mistake only affects this figure, as none of the other analyses disaggregated the base flow of the models.

*More generally, I think this points to a need to clearly map model fluxes and states onto the names used here. Which fluxes in each model are considered baseflow? Which states in each model are seen as soil moisture? Etc. A simple table should suffice.*

We agree that the inclusion of a table such as the one suggested could be very helpful to readers. We have included the following table in the supplementary material:

Table S2: Equivalence between hydrological fluxes/states and the model's outputs.

| Hydrological flux/state | Units | Hydrological model output used as a proxy | | | |
|---|---|---|---|---|---|
| | | GR4J | GR5J | GR6J | TUW |
| Runoff (Q) | mm/d | Qsim | Qsim | Qsim | q |
| Baseflow (BF) | mm/d | QR | QR | QR + QRExp | q2 |
| Actual evapotranspiration (ET) | mm/d | AE | AE | AE | eta |
| Snowmelt | mm/d | Melt | Melt | Melt | melt |
| Snow water equivalent (SWE) | mm | SnowPack | SnowPack | SnowPack | swe |
| Soil moisture (SM) | % | Prod/X1 | Prod/X1 | Prod/X1 | moist/FC |

The notation in the table corresponds to that used in the R packages of the models. Therefore, we also include a brief description of the model outputs, which are used as a proxy to study the hydrological fluxes/states of interest and references to the corresponding R packages.

*Caption Figure 9: I don't think either of these have been defined yet.*

Both rapid and seasonal transitions have been defined in L159 in the pre-print. However, to avoid confusion, the corresponding definition is included in parentheses. Thus, the new caption reads as follows:

Figure 9. Results of the analysis of variance (ANOVA) applied to the Critical Success Index (CSI) for droughts, floods, all drought-flood transitions (i.e., rapid and seasonal), rapid transitions (<14 days), and seasonal transitions (<90 days).

*L403-406: This is an interesting observation, but what explains this? Are the models just much worse in these landscapes and therefore bad at predicting extremes; is the daily time step too coarse to predict rapid events; something else? Adding some explanation for these findings would take this section beyond merely descriptive to some sort of understanding.*

We agree that this point could be further explored. We have included some thoughts on this in the discussion.

*L414-L415: That we need good forcing is of course true, but an alternative explanation of the importance of these particular parameters is that the calibration routine uses them to compensate for some sort of model deficiency by just changing the inputs. I think this alternative explanation deserves at least a mention too.*

We agree on this point and have incorporated this into the discussion, specifically in Section 5.3 (part of interest underlined):

Given the relative importance shown by the forcing adjustment parameters (i.e., dP and dT for all the models as well as SCF in TUW model, which seeks to correct the snow undercatch; Figure 11), the meteorological forcings can also have a major impact for detecting streamflow extremes and their transitions. Several studies have shown that errors in meteorological forcing are a key challenge in hydrological modeling (e.g., Brunner, 2023; Döll et al., 2016) due to, e.g., their large influence on the simulation of snow processes (e.g., Tang et al., 2023; Günther et al., 2019), or significant impacts on the partitioning between evaporation and runoff (e.g., Nasonova et al., 2011). Here, we attempt to reduce this effect by (1) utilizing local meteorological products over global ones, based on the evidence that these may enhance hydrological modeling (e.g., Clerc-Schwarzenbach et al., 2024), and (2) incorporating adjustment factors to account for potential systematic biases associated with them (e.g., Hughes, 2024; Probst and Mauser, 2022). However, introducing forcing adjustment factors could compensate for some model deficiencies by modifying the inputs (e.g., Tang et al., 2023, 2025). This is somehow reflected by the high dispersion of forcing adjustment factors within each configuration (Figure S16 in the Supplementary Material). Therefore, an improvement

in the spatiotemporal representation of precipitation and temperature, as well as of the potential interactions between these variables, could contribute to improved representations of compound streamflow extreme events in hydrological models.

*L440: These terms need to be defined.*

Both rapid and seasonal transitions have been defined in L159 in the pre-print. However, to avoid confusion, the corresponding definition is included in parentheses. The sentence now reads as follows:

This is consistent with the poor performance of all the models tested in capturing rapid transitions (i.e., occurring within 14 days; median CSI equal to zero when rapid and seasonal transitions are analyzed separately; not shown).

*L447-L448: This needs a mention/discussion of Mizukami et al., 2019, as well.*

These results are compared with the work of Mizukami et al. (2019) in the discussion section of the manuscript, where the following can be read:

Our comparison also highlights that the potential benefit from adjusting these choices (e.g., using other weights or other transformations) varies widely between catchments (Figure 5). This is in line with the findings of Mizukami et al. (2019), who found that the influence of weights on model performance depends on model structure and catchment characteristics.

*L518-L520: So it's not so much that the models improve at detecting droughts (as line 516 states) but that there are fewer opportunities for the model to fail? Maybe that should be the key point instead of the more optimistic phrasing that the earlier sentence uses.*

We agree. We have rewritten the sentence presented in L516 as follows:

The results of this sensitivity analysis indicate that using more flexible thresholds to define droughts (i.e., higher percentiles) can enhance the detection of these events, as more instances are identified, and they tend to be less severe compared to more restrictive thresholds.

*L523-L524: For GR4J and HBV I think it can be said that they are widely used, but despite being heavily inspired by it TUW is not HBV and I don't think there are quite as many studies with GR5J and GR6J out there as there are ones with GR4J. Also, it could be argued that the GR-suite of models are more different flavors of one model than three clearly different individual models. I'd thus suggest to add some nuance to this highlighted sentence.*

We have rewritten the sentence as follows:

To support our analysis, we tested four bucket-type hydrological models used within the hydrological modeling community (Addor and Melsen, 2019). Even though these models are at the lower end in terms of model complexity (Hrachowitz and Clark, 2017), and three of them share the same core structure, they allowed us to perform a comprehensive analysis of different model structures at a lower computational cost than when using models with more complex structures (e.g., Clark et al., 2017; Orth et al., 2015; Poncelet et al., 2017). Furthermore, previous studies have also shown that more complexity does not necessarily imply better performance (Figure 7; e.g., Li et al., 2015; Merz et al., 2022).

*L534: Not really the most appropriate references for this, I think. At the very least one of the early Kratzert papers should be added here.*

We have updated the references included. We are now using as examples the studies carried out by Kratzert et al. (2018), Frame et al. (2022), and Acuña Espinoza et al. (2025), which we considered are more suitable to make the point we want to highlight in the sentence. The part of the paragraph now reads as follows (underlining the point associated with the comment):

Our results provide insights on possible avenues of future research that could benefit drought-to-flood transitions modeling, which include: (1) exploring the use of modular platforms and a multi-model ensemble approach to quantify model uncertainty and identify more suitable model structures (e.g., Saavedra et al., 2022); (2) improving our understanding of the role of the spatial variability of precipitation for accurate flood simulations (e.g., Macdonald et al., 2025; Astagneau et al., 2022); (3) assessing the benefits of model runs at a subdaily timestep (e.g., hourly); and (4) exploring alternative data-driven modeling approaches such as long short-term memory (LSTM) networks (e.g., Frame et al., 2022; Acuña Espinoza et al., 2025; Kratzert et al., 2018).

*L545: This too general. Even though this study tests a wide variety of things, looking at 4 models (3 of which mostly differ in one or two equations) is not enough to make so bold of a claim. This sentence should read: "Drought events are better captured by the four hydrological models that we tested than flood events."*

We agree. However, to narrow our study, we have removed this conclusion from the take-home messages.

*L547: I think a conclusion section should be readable on its own, and right now it's not particularly clear what the third event type is. Transitions, I assume?*

In fact, in the sentence "In addition, model performance with respect to capturing all three event types varies widely between catchments.", associated with the first conclusion, transitions were referred to as the third type of event. However, to narrow our study, we have removed this conclusion from the take-home messages.

*L557: Might be good to add a few words to describe what this means.*

To narrow our study, we have removed this conclusion from the take-home messages.

*L563-L565: I don't think this is so clear cut, as mentioned before. These parameters also give the calibration routine much more freedom to make the models match the streamflow signal by any means necessary (i.e., a potential to get the right results for the wrong reasons). I don't think it's wrong to mention that accurate forcing data is helpful, but I don't think this can be so cleanly concluded from this particular experiment (nor do I think this is such a novel thought that it should be a bullet point here - the idea of "garbage in, garbage out" has existed for a long time).*

We agree. To narrow our study, we have removed this conclusion from the take-home messages. However, we included the following reflections about this point in the discussion section (limitations and future work):

Given the relative importance shown by the forcing adjustment parameters (i.e., dP and dT for all the models as well as SCF in TUW model, which seeks to correct the snow undercatch; Figure 11), the meteorological forcings can also have a major impact for detecting streamflow extremes and their transitions. Several studies have shown that errors in meteorological forcing are a key challenge in hydrological modeling (e.g., Brunner, 2023; Döll et al., 2016) due to, e.g., their large influence on the simulation of snow processes (e.g., Tang et al., 2023; Günther et al., 2019), or significant impacts on the partitioning between evaporation and runoff (e.g., Nasonova et al., 2011). Here, we attempt to reduce this effect by (1) utilizing local meteorological products over global ones, based on the evidence that these may enhance hydrological modeling (e.g., Clerc-Schwarzenbach et al., 2024), and (2) incorporating adjustment factors to account for potential systematic biases associated with them (e.g., Hughes, 2024; Probst and Mauser, 2022). However,

introducing forcing adjustment factors could compensate for some model deficiencies by modifying the inputs (e.g., Tang et al., 2023, 2025). This is somehow reflected by the high dispersion of forcing adjustment factors within each configuration (Figure S16 in the Supplementary Material). Therefore, an improvement in the spatiotemporal representation of precipitation and temperature, as well as of the potential interactions between these variables, could contribute to improved representations of compound streamflow extreme events in hydrological models.

*L567: This may also be too general. There is a lot of literature out there suggesting that one of the main flood generating mechanisms is snowmelt. Therefore in regions where this is the case, getting accurate SWE will be important. It is possible that this is not the case in the basins used in this work, but that should be indicated as such: "Our models' performance in simulating streamflow extremes and transitions in this sample of X basins depends .."*

We understand the reviewer's point, and from a critical standpoint, our statement may be somewhat confusing and should be modified. Logically, timing will be closely dependent on, e.g., the modeling of snowpack accumulation/melting processes in snow-dominated regions, but in our analysis, the highest relative importance was captured by streamflow timing (which implicitly includes this component). We have rewritten the statement as follows:

A model's performance in simulating streamflow extremes and transitions primarily depends on how well it captures streamflow timing rather than other hydrological signatures or variables such as evapotranspiration or snow water equivalent. However, streamflow timing is directly linked to, e.g., snowpack accumulation/melting processes, precipitation phase, and evapotranspiration processes. Consequently, verification of the dynamics of these processes can complement the diagnosis of model performance in capturing streamflow timing and the catchments' responsiveness.

*L582: I'd suggest to simply put it on Github, create a release, and assign a DOI to that. This is good practice, in line with the HESS recommendations for authors, and straightforward: https://docs.github.com/en/repositories/archiving-a-github-repository/referencing-and-citing-content*

The R-scripts used to identify events, calibrate models, and analyze results will be included in the database created as part of the study.

**References**

Acuña Espinoza, E., Loritz, R., Kratzert, F., Klotz, D., Gauch, M., Álvarez Chaves, M., and Ehret, U.: Analyzing the generalization capabilities of a hybrid hydrological model for extrapolation to extreme events, Hydrology and Earth System Sciences, 29, 1277–1294, https://doi.org/10.5194/hess-29-1277-2025, 2025.

Alvarez-Garreton, C., Mendoza, P. A., Boisier, J. P., Addor, N., Galleguillos, M., Zambrano-Bigiarini, M., Lara, A., Puelma, C., Cortes, G., Garreaud, R., McPhee, J., and Ayala, A.: The CAMELS-CL dataset: catchment attributes and meteorology for large sample studies – Chile dataset, Hydrology and Earth System Sciences, 22, 5817–5846, https://doi.org/10.5194/hess-22-5817-2018, 2018.

Araya, D., Mendoza, P. A., Muñoz-Castro, E., and McPhee, J.: Towards robust seasonal streamflow forecasts in mountainous catchments: impact of calibration metric selection in hydrological modeling, Hydrology and Earth System Sciences, 27, 4385–4408, https://doi.org/10.5194/hess-27-4385-2023, 2023.

Ayala, Á., Farías-Barahona, D., Huss, M., Pellicciotti, F., McPhee, J., and Farinotti, D.: Glacier runoff variations since 1955 in the Maipo River basin, in the semiarid Andes of central Chile, The Cryosphere, 14, 2005–2027, https://doi.org/10.5194/tc-14-2005-2020, 2020.

Berghuijs, W. R., Sivapalan, M., Woods, R. A., and Savenije, H. H. G.: Patterns of similarity of seasonal water balances: A window into streamflow variability over a range of time scales, Water Resources Research, 50, 5638–5661, https://doi.org/10.1002/2014WR015692, 2014.

Brunner, M. I., Anderson, B., and Muñoz-Castro, E.: Meteorological and hydrological dry-to-wet transition events are only weakly related over European catchments, Environ. Res. Lett., 20, 084013, https://doi.org/10.1088/1748-9326/ade72c, 2025.

Carrasco-Escaff, T., Rojas, M., Garreaud, R. D., Bozkurt, D., and Schaefer, M.: Climatic control of the surface mass balance of the Patagonian Icefields, The Cryosphere, 17, 1127–1149, https://doi.org/10.5194/tc-17-1127-2023, 2023.

Cinkus, G., Mazzilli, N., Jourde, H., Wunsch, A., Liesch, T., Ravbar, N., Chen, Z., and Goldscheider, N.: When best is the enemy of good – critical evaluation of performance criteria in hydrological models, Hydrology and Earth System Sciences, 27, 2397–2411, https://doi.org/10.5194/hess-27-2397-2023, 2023.

Frame, J. M., Kratzert, F., Klotz, D., Gauch, M., Shalev, G., Gilon, O., Qualls, L. M., Gupta, H. V., and Nearing, G. S.: Deep learning rainfall–runoff predictions of extreme events, Hydrology and Earth System Sciences, 26, 3377–3392, https://doi.org/10.5194/hess-26-3377-2022, 2022.

Garcia, F., Folton, N., and Oudin, L.: Which objective function to calibrate rainfall–runoff models for low-flow index simulations?, Hydrological Sciences Journal, 62, 1149–1166, https://doi.org/10.1080/02626667.2017.1308511, 2017.

Hammond, J., Anderson, B., Simeone, C., Brunner, M., Muñoz-Castro, E., Archfield, S., Magee, E., and Armitage, R.: Hydrological Whiplash: Highlighting the Need for Better Understanding and Quantification of Sub-Seasonal Hydrological Extreme Transitions, Hydrological Processes, 39, e70113, https://doi.org/10.1002/hyp.70113, 2025.

Knoben, W. J. M.: Setting expectations for hydrologic model performance with an ensemble of simple benchmarks, Hydrological Processes, 38, e15288, https://doi.org/10.1002/hyp.15288, 2024.

Knoben, W. J. M., Freer, J. E., Peel, M. C., Fowler, K. J. A., and Woods, R. A.: A Brief Analysis of Conceptual Model Structure Uncertainty Using 36 Models and 559 Catchments, Water Resources Research, 56, e2019WR025975, https://doi.org/10.1029/2019WR025975, 2020.

Kratzert, F., Klotz, D., Brenner, C., Schulz, K., and Herrnegger, M.: Rainfall–runoff modelling using Long Short-Term Memory (LSTM) networks, Hydrology and Earth System Sciences, 22, 6005–6022, https://doi.org/10.5194/hess-22-6005-2018, 2018.

Lema, F., Mendoza, P. A., Vásquez, N. A., Mizukami, N., Zambrano-Bigiarini, M., and Vargas, X.: Technical note: What does the Standardized Streamflow Index actually reflect? Insights and implications for hydrological drought analysis, Hydrology and Earth System Sciences, 29, 1981–2002, https://doi.org/10.5194/hess-29-1981-2025, 2025.

Melsen, L. A., Puy, A., Torfs, P. J. J. F., and Saltelli, A.: The rise of the Nash-Sutcliffe efficiency in hydrology, Hydrological Sciences Journal, 70, 1248–1259, https://doi.org/10.1080/02626667.2025.2475105, 2025.

Muñoz-Castro, E., Mendoza, P. A., Vásquez, N., and Vargas, X.: Exploring parameter (dis)agreement due to calibration metric selection in conceptual rainfall–runoff models, Hydrological Sciences Journal, 68, 1754–1768, https://doi.org/10.1080/02626667.2023.2231434, 2023.

Pizarro, A. and Jorquera, J.: Advancing objective functions in hydrological modelling: Integrating knowable moments for improved simulation accuracy, Journal of Hydrology, 634, 131071, https://doi.org/10.1016/j.jhydrol.2024.131071, 2024.

Pushpalatha, R., Perrin, C., Le Moine, N., Mathevet, T., and Andréassian, V.: A downward structural sensitivity analysis of hydrological models to improve low-flow simulation, Journal of Hydrology, 411, 66–76, https://doi.org/10.1016/j.jhydrol.2011.09.034, 2011.

Pushpalatha, R., Perrin, C., Moine, N. L., and Andréassian, V.: A review of efficiency criteria suitable for evaluating low-flow simulations, Journal of Hydrology, 420–421, 171–182, https://doi.org/10.1016/j.jhydrol.2011.11.055, 2012.

Seibert, J., Vis, M. J. P., Lewis, E., and van Meerveld, H. j.: Upper and lower benchmarks in hydrological modelling, Hydrological Processes, 32, 1120–1125, https://doi.org/10.1002/hyp.11476, 2018.

Swain, D. L., Prein, A. F., Abatzoglou, J. T., Albano, C. M., Brunner, M., Diffenbaugh, N. S., Singh, D., Skinner, C. B., and Touma, D.: Hydroclimate volatility on a warming Earth, Nat Rev Earth Environ, 6, 35–50, https://doi.org/10.1038/s43017-024-00624-z, 2025.

Woods, R. A.: Analytical model of seasonal climate impacts on snow hydrology: Continuous snowpacks, Advances in Water Resources, 32, 1465–1481, https://doi.org/10.1016/j.advwatres.2009.06.011, 2009.

Zhao, F., Nie, N., Liu, Y., Yi, C., Guillaumot, L., Wada, Y., Burek, P., Smilovic, M., Frieler, K., Buechner, M., Schewe, J., and Gosling, S. N.: Benefits of Calibrating a Global Hydrological Model for Regional Analyses of Flood and Drought Projections: A Case Study of the Yangtze River Basin, Water Resources Research, 61, e2024WR037153, https://doi.org/10.1029/2024WR037153, 2025.